# Novel Selenoesters as a Potential Tool in Triple-Negative Breast Cancer Treatment

**DOI:** 10.3390/cancers14174304

**Published:** 2022-09-02

**Authors:** Dominika Radomska, Robert Czarnomysy, Anna Szymanowska, Dominik Radomski, Enrique Domínguez-Álvarez, Anna Bielawska, Krzysztof Bielawski

**Affiliations:** 1Department of Synthesis and Technology of Drugs, Medical University of Bialystok, Kilinskiego 1, 15-089 Bialystok, Poland; 2Department of Biotechnology, Medical University of Bialystok, Kilinskiego 1, 15-089 Bialystok, Poland; 3Instituto de Química Orgánica General (IQOG-CSIC), Consejo Superior de Investigaciones Científicas, Juan de la Cierva 3, 28006 Madrid, Spain

**Keywords:** breast cancer, triple-negative breast cancer, anticancer drugs, selenium compounds, organoselenium compounds, selenoesters, apoptosis, cell signaling, flow cytometry

## Abstract

**Simple Summary:**

Breast cancer ranks at the forefront of all malignancies worldwide, and the numerous incidences and mortality rate associated with it are very burdensome to the health care system. Consequently, there is a constant need for new effective drugs with anticancer activity. There have been reports of highly cytotoxic effects of selenium compounds against cancer cells for some time. Hence, our team decided to evaluate the anticancer activity of novel selenoesters in MCF-7 and MDA-MB-231 breast cancer cells. Our results reveal that these compounds are cytotoxic at very low micromolar concentrations, which is associated with the induction of apoptosis and autophagy and arrest of breast cancer cells in the S or G_2_/M phase of the cell cycle. The obtained results are promising and show that selenium-containing compounds are worth more attention, as they show considerable potential for future candidates as anticancer agents.

**Abstract:**

Disturbing cancer statistics, especially for breast cancer, are becoming a rationale for the development of new anticancer therapies. For the past several years, studies have been proving a greater role of selenium in the chemoprevention of many cancers than previously considered; hence, a trend to develop compounds containing this element as potential agents with anticancer activity has been set for some time. Therefore, our study aimed to evaluate the anticancer activity of novel selenoesters (EDA-71, E-NS-4) in MCF-7 and MDA-MB-231 human breast cancer cells. The assays evaluating proliferation and cell viability, and flow cytometer analysis of apoptosis/autophagy induction, changes in mitochondrial membrane potential, disruption of cell cycle phases, and protein activity of mTOR, NF-κB, cyclin E1/A2, and caspases 3/7, 8, 9, 10 were performed. The obtained results indicate that the tested selenoesters are highly cytotoxic and exhibit antiproliferative activity at low micromolar doses (<5 µM) compared with cisplatin. The most active compound—EDA-71—highly induces apoptosis, which proceeds via both pathways, as evidenced by the activation of all tested caspases. Furthermore, we observed the occurrence of autophagy (↓ mTOR levels) and cell cycle arrest in the S or G_2_/M phase (↓ cyclin E1, ↑ cyclin A2).

## 1. Introduction

Breast cancer (BC) is one of the most commonly diagnosed cancers, which is a very significant clinical problem. It is estimated that there were nearly 2,300,000 new cases of this cancer in 2020, which is around 6300 new diagnoses per day. Unfortunately, these numbers are intimidating and treatment is becoming more and more challenging for clinicians due to increasing chemoresistance and the high mortality rate associated with it [1]. For this reason, there is a constant need for improving methods involving early cancer diagnostics or creating new drugs with precise molecular action and high anticancer activity, including overcoming multidrug resistance (MDR).

Oncologic surgery is the main therapeutic procedure for the treatment of BC, and systemic administration of cytostatic drugs is based on a strategy of neoadjuvant therapy (before surgery, to support surgical treatment), adjuvant therapy (after surgery, as an adjunct to operation), or induction therapy (to reduce tumor mass so that surgery can be performed) [2]. In 1978, the U.S. Food and Drug Administration (FDA) approved cisplatin for oncology treatment and today it is one of the most commonly used drugs during chemotherapy, especially in triple-negative BC (TNBC). Therapy with this agent is burdened with many harsh side effects that limit its effective dose and clinical use, but this is not the most serious medical problem [3]. The major difficulty is the increasing MDR of cancer cells to this substance [4]. From the available evidence, long-term treatment of BC with cisplatin leads to the development of resistance and a decrease in its therapeutic efficacy, despite the initial inhibition of tumor growth [5]. Therefore, based on these two premises, the search for anticancer agents with a high therapeutic index, exhibiting the potential to overcome MDR, is ongoing.

Selenium (Se) belongs to the group of trace elements. Despite its relatively small quantities in the human body, it exerts many important functions in it. It manifests its activity mainly through selenoproteins, into which it is incorporated in the form of selenomethionine (SeMet) or selenocysteine (Sec). These selenoamino acids are found in two important antioxidant enzymes—glutathione peroxidase (GPx) and thioredoxin reductase (TrxR)—which protect cells from oxidative stress. Moreover, Se is involved in the metabolism of thyroid hormones (it is incorporated into the enzyme—iodothyronine deiodinase (DIO)) and the recovery of vitamin C and E from their metabolites. Its indispensable role is also seen in the aging process, fertility, and the immune system, and recent reports indicate its preventive effect against cancer [6,7,8]. Meta-analyses and systematic reviews of clinical trials with Se confirm its chemopreventive activity in lung [9], thyroid [10], and breast cancer [11], among others. This effect is probably due to not only the neutralization of free radicals but also the inhibition of neoangiogenesis and induction of apoptosis in cancer cells [6], which would indicate that compounds containing Se in their structure could be potential chemopreventive or anticancer agents.

There are many Se compounds among which there are two main chemical groups: inorganic and organic derivatives. Inorganic Se compounds—selenites, selenates—exhibit genotoxic effects; hence, organic Se-containing compounds devoid of this toxicity to DNA have gained more attention. The group of organoselenium compounds consists of more than 10 chemical classes—including selenoesters [12]. These compounds are characterized by rapid action as a result of their molecule hydrolysis in the intracellular environment and release of reactive Se-containing forms with anticancer activity [13]. In the course of research performed by many teams all over the world, it was shown that selenoesters possess high cytotoxic potential, even in nanomolar concentrations, against many types of cancers (lung, colorectal, hepatic, pancreatic, ovarian, cervical, skin) [14,15,16,17]. Moreover, they exhibit the ability to overcome MDR [18,19,20], which provides a promising premise for further studies on this class of compounds and to explore their molecular mechanism of anticancer effects. Therefore, the aim of our study was an in-depth evaluation of the anticancer activity of novel selenoesters (EDA-71, E-NS-4) in MCF-7 and MDA-MB-231 human breast cancer cells.

## 2. Materials and Methods

### 2.1. Materials

Cisplatin, 3-(4,5-dimethylthiazol-2-yl)-2,5-diphenyltetrazolium bromide (MTT), dimethyl sulfoxide (DMSO), formaldehyde, glycine, methanol, sodium hydroxide, sodium dodecyl sulfate (SDS), and Tris were purchased from Sigma-Aldrich (St. Louis, MO, USA). Ethanol and sodium chloride were obtained from Avantor Performance Materials (Poland), while hydrochloric acid and trichloroacetic acid (TCA) were from Chempur (Poland). Stock cultures of human breast cancer cells (MCF-7 and MDA-MB-231) and normal human breast epithelial cells (MCF-10A) were provided by the American Type Culture Collection (ATCC, Manassas, VA, USA). Dulbecco’s Minimal Eagle Medium (DMEM), fetal bovine serum (FBS), phosphate-buffered saline (PBS) used in a cell culture, trypsin, glutamine, penicillin, and streptomycin were from Gibco (San Diego, CA, USA). An MEGM Mammary Epithelial Cell Growth Medium BulletKit was purchased from Lonza Bioscience (Basel, Switzerland). [^3^H]-thymidine (7 Ci/mmol) was received from Moravek Biochemicals (Brea, CA, USA), and Scintillation Coctail Ultima Gold XR from PerkinElmer (Waltham, MA, USA). DNase-free RNase A Solution was a product of Promega (Madison, WI, USA). FITC Annexin V Apoptosis Detection Kit II, JC-1 MitoScreen Kit and Stain Buffer were from BD Pharmigen (San Diego, CA, USA). An Autophagy Assay kit, FAM-FLICA^®^ Caspase-3/7 Assay kit, FAM-FLICA^®^ Caspase-8 Assay kit, FAM-FLICA^®^ Caspase-9 Assay kit, FAM-FLICA^®^ Caspase-10 Assay kit, and propidium iodide were purchased from ImmunoChemistry Technologies (Bloomington, MN, USA). Cyclin E1 Mouse mAb, Cyclin A2 Rabbit mAb, mTOR Rabbit mAb (Alexa Fluor^®^ 647 Conjugate), NF-κB p65 XP^®^ Rabbit mAb (Alexa Fluor^®^ 647 Conjugate), Alexa Fluor^®^ 647-labeled secondary anti-mouse antibody, Alexa Fluor^®^ 488-labeled secondary anti-rabbit antibody were products of Cell Signaling Technology (Beverly, MA, USA).

### 2.2. Tested Compounds

The two tested compounds (Figure 1) are EDA-71 (Se-(2-oxopropyl) 4-chlorobenzoselenoate) and E-NS-4 (Se-cyanomethyl 4-chlorobenzoselenoate). They share the selenoester functional group (-(C=O)-Se-) and the aryl moiety (4-chlorophenyl) bound to the carbonyl of the selenoester. The difference lies in the alkyl moiety bound to the selenium atom, which contains a ketone group in EDA-71 (2-oxopropyl) and a nitrile group in E-NS-4 (2-cyanomethyl). The synthesis and characterization of the two selenocompounds have been described in previous works: EDA-71 in [14] and E-NS-4 in [21]. Briefly, 4-chlorobenzoyl chloride is added over a solution of sodium hydrogen selenide generated *in situ* by the reduction of elemental grey selenium with sodium borohydride. After completion of the reaction, the crude of reaction is filtered to remove the salts generated in the process, and over the filtrate, chloroacetone or chloroacetonitrile is added, to form EDA-71 and E-NS-4, respectively. The desired compound precipitates from the reaction media and it is filtered, dried, and purified as previously described [14,21]. 

### 2.3. Cell Culture of MCF-7, MDA-MB-231, and MCF-10A Cells

Human breast cancer cell lines (MCF-7 and MDA-MB-231) and normal human breast epithelial cells (MCF-10A) were purchased from the American Type Culture Collection (ATCC, Manassas, VA, USA). MCF-7 and MDA-MB-231 cells were cultured in Dulbecco’s Modified Eagle Medium (Gibco, San Diego, CA, USA), MCF-10A cells were cultured in Mammary Epithelial Cell Growth Medium with supplements: BPE, hEGF, insulin, hydrocortisone, GA-1000 (Lonza, Basel, Switzerland). All media were complemented by 10% of fetal bovine serum (FBS) and 1% of antibiotics: penicillin and streptomycin (both Gibco, San Diego, CA, USA). The cells were maintained in an incubator that provides the optimal growth conditions for the cell culture: 5% CO_2_, 37 °C, and humidity in a range of 90–95%. The cells were cultured in 100 mm plates (Sarstedt, Newton, NC, USA). Subsequently after obtaining a subconfluent cell culture, the cells were detached with 0.05% trypsin with 0.02% EDTA (Gibco, San Diego, CA, USA). Then, utilizing a Scepter 3.0 handheld automated cell counter (Milipore, Burlington, MA, USA), the number of cells was quantified and seeded at a density of 5 × 10^5^ cells per well in six-well plates (“Nunc”) in 2 mL of the growth medium (Dulbecco’s Modified Eagle Medium or Mammary Epithelial Cell Growth Medium, respectively). In the present study, cells that obtained 80% of confluence were used.

### 2.4. Cell Viability Assay

The cytotoxic activity of the tested compounds against breast cancer (MCF-7 and MDA-MB-231) and normal human breast epithelial (MCF-10A) cells was determined by MTT assay according to Carmichael’s method. The evaluation of cell viability was based on measuring their ability to enzymatically reduce yellow tetrazolium salt (3-(4,5-dimethylthiazol-2-yl)-2,5-diphenyltetrazolium bromide, MTT) to purple formazan under the influence of the enzyme—mitochondrial dehydrogenase, which is only found in living cells [22]. All cultured cell lines were treated with different concentrations of the tested compounds in medium (0.5, 1, 1.5, 2, 2.5, 3, 3.5, 4, 4.5, and 5 µM) and incubated for 24 h in an incubator providing optimal conditions for cell culture growth (37 °C temperature, 5% CO_2_, and 90–95% humidity) in six-well plates (baseline seeding density: 5 × 10^5^ cells/well). After this time, the medium was removed and the cell monolayer was washed with warm phosphate-buffered saline (PBS) without calcium and magnesium. In the next step, PBS and MTT solution in PBS (5 mg/mL) were added to each well to obtain a final MTT concentration of 0.5 mg/mL, and then the cells were incubated for 4 h (37 °C temperature, 5% CO_2_, 90–95% humidity). After the incubation period, the content of each well was removed, 1 mL of DMSO was added to dissolve the formazan, and then the plate with the added DMSO was mixed for 5–10 min on a microplate shaker (Boeco, Hamburg, Germany). Then, 10 µL of Sorensen’s glycine buffer (a solution of 0.1 M glycine and 0.1 M sodium chloride adjusted to pH 10.5 with 0.1 M sodium hydroxide) was added and the absorbance of the obtained solution was immediately measured at a wavelength of λ = 570 nm using a Thermo Scientific Evolution 201 UV-VIS spectrophotometer (Thermo Fisher Scientific, Waltham, MA, USA). The absorbance result obtained in the control cells (without the addition of the tested compounds) was taken as 100%, while the viability of cells incubated with the tested compounds was presented as a percentage of the control value.

### 2.5. [^3^H]-Thymidine Incorporation Assay

The antiproliferative activity of the tested compounds was evaluated by measuring the incorporation of [^3^H]-thymidine into the cancer cell DNA after 24 h of incubation. MCF-7 and MDA-MB-231 breast cancer cells were seeded into six-well plates (density 5 ×10^5^ cells/well) and cultured for 24 h in an incubator under optimal conditions for cell growth (37 °C, 5% CO_2_, 90–95% humidity). Then, they were incubated under the same conditions for 24 h with a growth medium containing the tested compounds at different concentrations (0.5, 1, 1.5, 2, 2.5, 3, 3.5, 4, 4.5, and 5 µM). After 24 h, the medium was removed, the cells were washed with PBS (Corning, Kennebunk, ME, USA), and a fresh medium was added. After that, the cells were treated with 0.5 µCi of tritium-labeled thymidine (specific activity 7 Ci/mmol) by incubating them with this compound for 4 h under the same conditions as the cell culture. In the first step, the medium was removed and the cell monolayer was washed twice with 1 mL of 0.05 M Tris-HCl buffer pH 7.4 containing 0.11 M NaCl. To denature proteins, the cells were washed twice with 1 mL of 5% trichloroacetic acid (TCA) solution. Finally, cell lysis was performed by adding 1 mL of 0.1 M sodium hydroxide (NaOH) solution containing 1% sodium dodecyl sulfate (SDS) to each well. After five minutes, the obtained cell lysates were transferred to scintillation vials with 2 mL of scintillation fluid added to them beforehand. Radioactivity was measured using the Scintillation Counter 1900 TR, TRI-CARB (Packard, Perkin Elmer, Inc., San Jose, CA, USA). The intensity of DNA biosynthesis in the analyzed cells was expressed as dpm/well. The result of radioactivity measurement in the control cells (without the addition of tested compounds) was taken as 100%, while the values of cells incubated with the tested compounds were presented as a percentage of the control value.

### 2.6. Flow Cytometry Assessment of Annexin V and Propidium Iodide Binding

The apoptosis induction by the tested compounds was assessed by the exposure of phosphatidylserine on the cell membrane, to which fluorescein isothiocyanate (FITC)-labeled annexin V binds with high affinity in the presence of Ca^2+^ calcium ions. The FITC Annexin V Apoptosis Detection Kit II (BD Pharmingen, San Diego, CA, USA) and a flow cytometer (BD FACSCanto II, BD Biosciences Systems, San Jose, CA, USA) were used for this detection. The assay was performed according to the manufacturer’s instructions. Breast cancer cells (MCF-7 and MDA-MB-231) were incubated for 24 h (37 °C, 5% CO_2_, 90–95% humidity) with the most active compound (EDA-71) and the reference drug (cisplatin) at concentrations of 1.5 and 3 µM. Flow cytometer calibration was performed by preparing two controls—a positive and a negative control. The positive control was cells in which apoptosis was induced using 2 µL of 3% formaldehyde in buffer and placing them on ice for 30 min. The negative control was cells that were not treated with any of the proapoptotic agents. First, in cells treated with the tested compounds as well as the controls, the medium was removed and the cells were washed twice with cold PBS. Subsequently, the cells were resuspended in Binding Buffer included in the kit at a concentration of 1 × 10^6^ cells/mL. From each sample, 100 µL of cell suspension was taken and transferred to test tubes, to which 5 µL each of FITC Annexin V and propidium iodide (PI) was then added. The contents of the test tubes were gently vortexed and incubated for 15 min at room temperature, protected from light. After the required time, the contents of the test tubes were made up to 500 µL with Binding Buffer and immediately analyzed in a flow cytometer (10,000 events measured). After the flow cytometer readout, the results were analyzed using FACSDiva software (BD Biosciences Systems, San Jose, CA, USA). The equipment was calibrated with BD Cytometer Setup and Tracking Beads (BD Biosciences, San Diego, CA, USA).

### 2.7. Assessment of Changes in Mitochondrial Membrane Potential

In the early phase of apoptotic death occurring via the intrinsic pathway, there is an increase in mitochondrial membrane permeability, resulting in a decrease in mitochondrial membrane potential (MMP, ΔΨ_m_). To detect these changes, the carbocyanine lipophilic cationic fluorochrome JC-1 included in the JC-1 MitoScreen kit (BD Pharmigen, San Diego, CA, USA) and a flow cytometer (BD FACSCanto II) with the appropriate software to analyze the obtained results (FACSDiva; both from BD Biosciences Systems) were used. The entire cell staining and cytometric analysis procedure was performed according to the instructions provided with the kit. MCF-7 and MDA-MB-231 cells were incubated for 24 h (37 °C, 5% CO_2_, 90–95% humidity) with cisplatin and compound EDA-71 (concentrations of 1.5 and 3 µM). After the incubation time with the tested compounds, the cells (1 × 10^6^ cells/sample) were washed and resuspended in 0.5 mL of buffer containing 10 µg/mL JC-1 dye. Incubation was carried out for 15 min at room temperature, protected from light. Afterwards, the cells were washed twice with buffer, resuspended in 0.5 mL PBS and immediately analyzed using a flow cytometer (BD FACSCanto II; 10,000 events measured) and FACSDiva software to count the percentage of cells with reduced ΔΨ_m_. The equipment was calibrated with BD Cytometer Setup and Tracking Beads (BD Biosciences, San Diego, CA, USA).

### 2.8. Caspases 3/7, 8, 9, and 10 Enzymatic Activity Assay

Activation of the caspase cascade occurs as a result of the initiation of the apoptotic process in the cell and is induced by the cytotoxic activity of the compound. In this regard, assessment of initiator (caspases 8, 9, and 10) and executioner (caspases 3 and 7) caspase activity was performed using FAM-FLICA^®^ Caspase Assay kits (all from ImmunoChemistry Technologies, Bloomington, MN, USA) according to the manufacturer’s instructions. After 24 h incubation of MCF-7 and MDA-MB-231, breast cancer cells with the tested compounds (concentrations of 1.5 and 3 µM), the cells were collected, washed twice with cold PBS, and resuspended in Apoptosis Wash Buffer to a final concentration of 5 × 10^5^ cells/mL. In the next step, 290 µL each of cell suspension was taken and transferred into tubes. Then, 10 µL of FLICA solution diluted immediately before use (1:5 *v*/*v*, using PBS) was added to the cells, mixed by pipetting, and incubated in the dark for 1 h at 37 °C. After this time, the cells were washed twice with 2 mL Apoptosis Wash Buffer, centrifuged, and resuspended in 300 µL of the buffer. Thus, prepared samples were immediately analyzed using a BD FACSCanto II flow cytometer (10,000 events) with FACSDiva software (both from BD Biosciences Systems, San Jose, CA, USA). The equipment calibration was performed using BD Cytometer Setup and Tracking Beads (BD Biosciences, San Diego, CA, USA).

### 2.9. Measuring the Number of Autophagosomes and Autolysosomes by Autophagy Assay

An autophagy assay was performed to evaluate whether the tested compounds induce the autophagy process in MCF-7 and MDA-MB-231 breast cancer cells. The stain included in the Autophagy Assay, Red kit (ImmunoChemistry Technologies, Bloomington, MN, USA) was an aliphatic molecule with the ability to enter the cell and fluoresce brightly upon binding to the lipid bilayer membranes of autophagosomes and autolysosomes. The entire assay was performed according to the manufacturer’s instructions. Cells were exposed to 1.5 and 3 µM of the tested compound (EDA-71) and cisplatin for 24 h. After drug treatment, unfixed cells were washed and resuspended in PBS at a concentration of 5 × 10^5^ cells/mL. Then, 490 µL each of cell suspension was taken, transferred to test tubes, and 10 µL of Autophagy Probe, Red solution (previously diluted 1:5 in PBS) was added and incubated (30 min, 37 °C, in the dark). After incubation, the cells were washed and resuspended in Cellular Assay Buffer, finally adding Fixative at a ratio of 1:5 (*v*/*v*). After this step, the prepared samples were immediately measured using a flow cytometer (BD FACSCanto II; 10,000 events measured), and the percentage of cells with an occurring autophagy process was calculated using FACSDiva software (both from BD Biosciences Systems, San Jose, CA, USA). The equipment calibration was performed using BD Cytometer Setup and Tracking Beads (BD Biosciences, San Diego, CA, USA).

### 2.10. Cell Cycle Analysis Using Flow Cytometry

Seeded at a density of 5 × 10^5^ cells/well (six-well plate) and then cultured for 24 h under optimal conditions for cell culture growth (37 °C, 5% CO_2_, 90–95% humidity), MCF-7 and MDA-MB-231 breast cancer cells were treated for 24 h with 1.5 and 3 µM EDA-71 and cisplatin. After 24 h, cells were collected and fixed in 1 mL of 70% ethanol. The thus prepared samples were placed in a freezer at -20 °C overnight. On the next day, the fixed cells were washed with cold PBS, centrifuged for 10 min at 2000 rpm, and the supernatant was removed. The cell pellet was then resuspended in PBS containing 50 μg/mL DNase-free RNase A Solution (Promega, Madison, WI, USA) and stained with 100 μg/mL PI (ImmunoChemistry Technologies, Bloomington, MN, USA). Incubation was performed for 30 min at 37 °C in the dark. In the next step, the cells were washed and resuspended in PBS. Cell cycle phase distribution was analyzed using the FACSCanto II flow cytometer (10,000 events measured) with FACSDiva software (both from BD Biosciences Systems, San Jose, CA, USA) and then FCS Express software (De Novo Software, Pasadena, CA, USA). The equipment was calibrated with BD Cytometer Setup and Tracking Beads (BD Biosciences, San Diego, CA, USA).

### 2.11. Determination of NF-κB and mTOR Using Flow Cytometry

The protein levels of NF-κB and mTOR in MCF-7 and MDA-MB-231 breast cancer cells after 24 h exposure to the tested compounds (EDA-71 and cisplatin; concentrations of 1.5 and 3 µM) were evaluated by flow cytometry. For this purpose, an mTOR antibody and an NF-κB antibody conjugated to Alexa Fluor^®^ 647 (all from Cell Signaling Technology, Beverly, MA, USA) were used. Each assay was performed according to the manufacturer’s protocol available on their website. After incubation with the tested compounds, the cells were centrifuged, resuspended in 4% formaldehyde (100 µL/1 × 10^6^ cells), and fixed for 15 min at room temperature. Afterwards, to remove the formaldehyde, the cells were washed by centrifugation with excess PBS and began the permeabilization step. For this purpose, ice-cold 90% methanol was added to the pre-chilled cell pellet, gently vortexed, and incubated for 1 h on ice. Later, the cells were washed by centrifugation with excess PBS again, resuspended in 100 µL of 1:100 diluted primary antibody (Stain Buffer was used for dilution), and incubated for 1 h at room temperature in the dark. After incubation, the cells were washed, resuspended in 300 µL PBS and immediately tested with a flow cytometer (BD FACSCanto II, BD Biosciences Systems, San Jose, CA, USA; 10,000 events measured). The obtained results were analyzed using FACSDiva software (BD Biosciences Systems, San Jose, CA, USA). The equipment calibration was performed using BD Cytometer Setup and Tracking Beads (BD Biosciences, San Diego, CA, USA).

### 2.12. Determination of Cyclin E1 and Cyclin A2 Using Flow Cytometry

Apart from analyzing the distribution of cell cycle phases, in this study, cyclins responsible for regulating the cell division cycle (cyclin E1 and cyclin A2) were also assayed. Up-regulation of cyclin E1 and its binding to a complex with cyclin-dependent kinase 2 (Cdk2) enables cell cycle transition from the G_1_ to the S phase, while cyclin A2/Cdk1 complex stimulates cell transition from the G_2_ to the M phase (mitosis). Identification of these proteins was performed after 24 h treatment of MCF-7 and MDA-MB-231 cells with the tested compounds (EDA-71 and cisplatin; concentrations of 1.5 and 3 µM). Cyclin E1 and cyclin A2 antibodies (both from Cell Signaling Technology, Beverly, MA, USA) were used in this study, and each assay was performed according to the manufacturer’s protocol available on their website. After incubation with the tested compounds, the cells were centrifuged, resuspended in 4% formaldehyde (100 µL/1 × 10^6^ cells), and fixed for 15 min at room temperature. Afterwards, to remove the formaldehyde, the cells were washed by centrifugation with excess PBS and began the permeabilization step. For this purpose, ice-cold 90% methanol was added to the pre-chilled cell pellet, gently vortexed, and incubated for 1 h on ice. Later, the cells were washed by centrifugation with excess PBS again, resuspended in 100 µL of 1:100 diluted primary antibody (Stain Buffer was used for dilution), and incubated for 1 h at room temperature in the dark. After the required incubation period, the cells were washed, resuspended in 100 µL of secondary antibody diluted in Stain Buffer (1:100 ratio; anti-mouse IgG Alexa Fluor^®^ 647 Conjugate in the case of cyclin E1 and anti-rabbit IgG Alexa Fluor^®^ 488 Conjugate in the case of cyclin A2; both from Cell Signaling Technology, Beverly, MA, USA) and incubated for 30 min at room temperature, protected from light. Subsequently, the cells were washed, resuspended in 300 µL PBS and immediately tested with a flow cytometer (BD FACSCanto II, BD Biosciences Systems, San Jose, CA, USA; 10,000 events measured). The obtained results were analyzed using FACSDiva software (BD Biosciences Systems, San Jose, CA, USA). The equipment calibration was performed using BD Cytometer Setup and Tracking Beads (BD Biosciences, San Diego, CA, USA).

### 2.13. Statistical Analysis

All results are presented as mean ± standard deviation (SD) from at least three independent experiments in triplicate. Statistical analysis was performed using GraphPad Prism 8 software (GraphPad Software, San Diego, CA, USA). Statistical differences between the experimental (treated cells) and control (untreated cells) groups were determined using one-way ANOVA followed by Dunnett’s test. Differences were considered statistically significant when *p* < 0.05.

## 3. Results

### 3.1. EDA-71 and E-NS-4 Suppress Growth and Exhibit High Cytotoxicity in Breast Cancer Cells

To evaluate *in vitro* cytotoxic activity of the tested selenium compounds—EDA-71 and E-NS-4 (Figure 1)—the viability of breast cancer cells (MCF-7 and MDA-MB-231) and normal human breast epithelial cells (MCF-10A) after the 24 h of exposure to them at different concentrations was measured by MTT assay. The reference compound in the present study was cisplatin. Both selenoesters exhibited high cytotoxicity against both breast cancer cell lines; however, compound EDA-71 was the most active (Figure 2). A similar finding was observed against normal human MCF-10A cells. Furthermore, the cytotoxic activity of the tested compounds was dose-dependent and—this deserves attention—was more potent against MDA-MB-231 triple-negative breast cancer cells. The IC_50_ of the most active compound (EDA-71) was 1.75 ± 0.06 µM and 1.40 ± 0.05 µM in MCF-7 and MDA-MB-231 cells, respectively. Meanwhile, the IC_50_ of E-NS-4 was 4.84 ± 0.15 µM in MCF-7 cells and 4.52 ± 0.36 µM in MDA-MB-231 cells. In turn, the IC_50_ values of EDA-71 and E-NS-4 in MCF-10A cells were 1.56 ± 0.05 µM and 4.36 ± 0.17 µM, respectively. For cisplatin, such strong cytotoxic activity was not observed, and the IC_50_ values for all evaluated cell lines were higher than the tested concentration range (>5.0 µM; Figure 2).

Considering the higher cytotoxic activity in breast cancer cells, compound EDA-71 (Figure 1) was selected for further studies on its effects on the molecular mechanism of apoptosis, autophagy, and cell cycle in MCF-7 and MDA-MB-231 cells.

DNA biosynthesis is a key process occurring during the proliferation of all cells in the body. Due to increased metabolism of cancer cells, it proceeds in an intensified and uncontrolled manner, causing rapid growth of pathological tissues. For this reason, compounds with anticancer activity should be characterized by antiproliferative effects. Therefore, to determine the mechanism responsible for the growth inhibition of MCF-7 and MDA-MB-231 breast cancer cells and MCF-10A normal human breast epithelial cells by novel selenoorganic compounds, the effect of these derivatives on DNA biosynthesis was investigated. In this study, the amount of incorporated radioactive [^3^H]-thymidine into the DNA of MCF-7, MDA-MB-231, and MCF-10A cells was measured after 24 h of incubation with the tested compounds and cisplatin at various concentrations. The obtained results reveal that cisplatin exhibited no significant effect on DNA biosynthesis, and the IC_50_ values for each cell line were higher than the tested concentration range (>5.0 µM; Figure 3). In contrast, the tested compounds were characterized by pronounced antiproliferative effects in normal breast epithelial cells and breast cancer cell lines in a dose-dependent manner. In MCF-7 cells, compounds EDA-71 and E-NS-4 exhibited similar growth inhibitory effects, with IC_50_ values of 2.44 ± 0.04 µM and 2.37 ± 0.04 µM, respectively. In contrast, in MDA-MB-231 cells, higher activity was observed with compound EDA-71 (IC_50_ = 1.55 ± 0.06 µM). Compound E-NS-4 had a more than 1.5-fold lower effect in the same cancer cell line, with an IC_50_ value of 2.49 ± 0.23 µM. For MCF-10A cells, the IC_50_ values were similar to those observed for breast cancer cells, at 2.18 ± 0.13 µM and 3.16 ± 0.11 µM for EDA-71 and E-NS-4, respectively. Importantly, EDA-71 again exhibited higher activity against MDA-MB-231 triple-negative breast cancer cells (Figure 3).

### 3.2. EDA-71 Induces Apoptosis via the Extrinsic and Intrinsic Pathway

Programmed cell death, or apoptosis, is a mechanism that maintains homeostasis in the body by eliminating pathological, dysfunctional, or damaged cells. However, cancer cells have the ability to evade this process, leading to the growth of abnormal tissue and tumor enlargement. For this reason, the used anticancer agents have to induce the apoptotic process in cancer cells, killing them [23]. With this objective, we evaluated the induction of the apoptotic process in MCF-7 and MDA-MB-231 breast cancer cells and MCF-10A normal human breast epithelial cells after 24 h of exposure to cisplatin and compound EDA-71 at two concentrations (1.5 and 3 µM). The main characteristic phenomenon observed during apoptosis is the externalization of phosphatidylserine (PS) on the cell membrane. Double staining with annexin V-FITC (AV) and propidium iodide (PI) is used to detect changes in the location of this phospholipid by flow cytometry and thus distinguish the determined cell populations. Viable cells are not stained with either of these reagents. Early and late apoptotic cells are stained with AV as it binds to the PS exposed on the cell membrane, whereas PI provides a way to study cell membrane integrity and to label late apoptotic and necrotic cells by staining cells with damaged cell membranes. In this regard, this assay allows for the determination of viable (AV^−^/PI^−^), early apoptotic (AV^+^/PI^−^), late apoptotic (AV^+^/PI^+^), and necrotic (AV^−^/PI^+^) cell populations. The results indicate that EDA-71 at a concentration of 1.5 µM induced apoptosis approximately twice as strong as cisplatin at the same concentration in both cell lines (Figure 4). In MCF-7 cells, 11.6 ± 1.2% and 22.5 ± 2.1% apoptotic cells (sum of early and late apoptotic cells) were observed for cisplatin and EDA-71 at a concentration of 1.5 µM, respectively, and 5.6 ± 0.7 and 11.1 ± 1.5% in MDA-MB-231 cells. For a higher concentration (3 µM) of the selenoester EDA-71, there were pronounced differences between the cell lines regarding the amount of cell populations in which apoptosis was induced. While in MCF-7 cells, a moderate increase in the number of apoptotic cells to 33.6 ± 2.8% was observed, in MDA-MB-231 cells the number of cells with induced apoptosis increased sharply to 61.8 ± 6.8% (a nearly 6-fold increase compared with the concentration of 1.5 µM). Furthermore, we observed that EDA-71 at a concentration of 3 µM induced necrosis in a part of the MDA-MB-231 cells (6.4 ± 1.7%). As for cisplatin (3 µM), it did not cause any major changes in the number of apoptotic cells, whose values were 15.0 ± 3.3% and 6.0 ± 1.2% for MCF-7 and MDA-MB-231 cells, respectively. A very interesting phenomenon observed during this study was that apoptosis induced by compound EDA-71 at a lower concentration (1.5 µM) was comparable between the two breast cancer cell lines, while at a higher concentration (3 µM) a more potent effect on MDA-MB-231 triple-negative breast cancer cells was observed (Figure 4). For normal MCF-10A cells, compound EDA-71 exhibited significantly higher proapoptotic properties than the reference compound. The percentage of apoptotic cells was 9.7 ± 1.7% and 10.5 1.3% for cisplatin at concentrations of 1.5 and 3 µM, respectively. In turn, EDA-71 caused an almost 4-fold (1.5 µM; 22.0 ± 1.3%) and more than 10-fold (3 µM; 70.6 ± 2.8%) increase in the percentage of apoptotic cells compared to the control group (6.5 ± 1.4%), indicating its considerable toxicity against normal cells. Moreover, as in the case of MDA-MB-231 cells, an increase in the necrotic cell population (4.9 ± 1.3%) was observed at a concentration of 3 µM of this tested compound (Figure 4C).

Apoptosis, and the crucial activation of initiator caspases in this process, may proceed in the cell via two pathways—extrinsic (death receptor-mediated) and intrinsic (mitochondria-mediated). In the extrinsic pathway, after stimulation of death receptors (e.g., Fas, TNFR 1/2, or DR4/5), recruitment of adaptor molecules, formation of death-inducing signaling complex (DISC), and then activation of caspases 8 and 10 occur. This, along with the subsequent activation of executory caspases, finally leads to apoptosis and cell death [23,24]. Moreover, there is also the interesting fact that caspase 10 mediates the sensitivity of cancer cells to TRAIL (tumor necrosis factor-related apoptosis-inducing ligand) induced apoptosis [25]. Considering the important role of caspases 8 and 10 in the initiation of apoptosis via the extrinsic pathway, we evaluated the effects of the novel selenocompound EDA-71 and cisplatin on the activation of these initiatory caspases in MCF-7 and MDA-MB-231 breast cancer cells after 24 h of exposure to these compounds at concentrations of 1.5 and 3 µM. We observed that the tested compound increased the amount of these caspases’ active forms in both breast cancer cell lines (Figure 5 and Figure 6). In MCF-7 cells, EDA-71 increased the population of cells with active caspase 8 more than 3.5-fold at a concentration of 1.5 µM and nearly 12-fold at a concentration of 3 µM compared with the control with values of 4.7 ± 0.3% and 15.5 ± 3.9%, respectively. Contrastingly, in MDA-MB-231 cells there was a nearly 1.5-fold (1.5 µM) and over 3.5-fold (3 µM) increase in caspase 8 activity following the effect of EDA-71 compared with the control. In this case, the values were 8.1 ± 0.6% and 21.6 ± 1.5%, respectively. In contrast, cisplatin did not significantly increase caspase 8 activity in both cell lines (Figure 5). For caspase 10, an approximately 1.5-fold increase in its activity was observed after exposure to 1.5 µM EDA-71 compared with the control in both breast cancer cell lines. There was 2.5 ± 0.5% and 7.7 ± 0.8% of cells with active caspase 10 in MCF-7 and MDA-MB-231 cells, respectively. At the higher concentration (3 µM) of the tested compound, an over 7-fold (10.0 ± 0.5%) in MCF-7 breast cancer cells and an approximately 3-fold (15.5 ± 2.7%) in MDA-MB-231 triple-negative breast cancer cells increase in caspase 10 activity compared with the control was reported. As for cisplatin, no significant effect on caspase 10 activation was found (Figure 6)—similar to that observed for caspase 8. The results for caspase 8 and 10 activities are consistent with those obtained in the AV/PI assay, indicating that apoptosis induced by the novel compound EDA-71 may proceed via an extrinsic pathway mediated by death receptors.

As previously mentioned, besides apoptosis induced by external stimulation of death receptors, it can also be initiated by the intrinsic pathway, in which mitochondria play a major role. Triggering factors for this pathway include DNA damage or oxidative stress, but there is also crosstalk with the extrinsic pathway mediated by caspase 8. It is known that activated caspase 8 cleaves the Bid (BH3 interacting-domain death agonist) protein resulting in the formation of an active protein called truncated Bid (tBid). This molecule and the proteins Bax and Bak incorporate into the mitochondrial membrane and oligomerize to form protein-permeable pores in this structure. The formation of macropores in the mitochondrial membrane causes its destabilization, an increase in permeability and a decrease in potential, and finally the release of cytochrome c and Ca^2+^ calcium ions into the cytosol of the cell [23,26]. To investigate the effect of the new selenoester on mitochondrial membrane potential (MMP, ∆Ψ_m_), the MCF-7 and MDA-MB-231 cells were treated with EDA-71 and cisplatin at different concentrations (1.5 and 3 µM) for 24 h, followed by cytometric analysis using JC-1 dye. JC-1 is a carbocyanine lipophilic cationic fluorochrome that can take on different forms and fluorescence depending on the MMP. In viable cells with normally functioning mitochondria, this dye accumulates in the hyperpolarized mitochondrial membrane to form aggregates that emit red fluorescence. In contrast, in apoptotic or necrotic cells, these aggregates disintegrate and monomers with green fluorescence are formed. As shown in Figure 7, the new compound caused an increase in the percentage of cells with decreased MMP in both breast cancer cell lines. In untreated cells (control group), the percentage of cells with depolarized mitochondria was 5.4 ± 1.1% and 11.4 ± 1.6% for MCF-7 and MDA-MB-231 cells, respectively. A lower concentration of EDA-71 (1.5 µM) led to a quite similar decrease in MMP in the tested breast cancer lines—it was 9.7 ± 0.7% (MCF-7) and 16.6 ± 0.3% (MDA-MB-231) of the cell population. Meanwhile, the concentration of 3 µM of the tested compound increased this percentage almost 6-fold in MCF-7 cells and almost 3-fold in MDA-MB-231 cells compared with the control group. In MCF-7 cells, 31.8 ± 0.7% of cells with decreased MMP were observed, while this percentage was 33.9 ± 1.5% in MDA-MB-231 cells. Cisplatin at a concentration of 1.5 µM—and also 3 µM in MDA-MB-231 cells—did not significantly affect the decrease in MMP, and the percentage of cells with depolarized mitochondria was comparable to the control (6.9 ± 0.9% in MCF-7; 11.4 ± 1.7% (1.5 µM) and 13.6 ± 0.4% (3 µM) in MDA-MB-231). Only the reference compound used at a concentration of 3 µM increased the number of MCF-7 cells with decreased mitochondrial potential, but the obtained values were lower than those found when EDA-71 was used at a concentration of 1.5 µM in the same cell line (Figure 7) and it was 7.6 ± 1.4%. Importantly, the results for MMP changes are consistent with those obtained in the AV/PI assay, highlighting that apoptosis induced by the novel compound EDA-71 may proceed via an intrinsic pathway mediated by mitochondria.

NF-κB is a protein transcription factor with antiapoptotic activity that is overexpressed in cancer cells [27]. Its antiapoptotic properties are due to, among other things, the induction of the expression of Bcl-2 family proteins that inhibit apoptosis. Recent reports indicate that blocking NF-κB may have a direct effect on apoptosis proceeding via an intrinsic pathway of which mitochondria are a key structure in the cell. It has been found that inhibition of NF-κB causes the release of cytochrome c from the mitochondria, initiating the apoptotic process [28]. The reason for this phenomenon could be explained in the fact that inhibition of this factor has a stimulatory effect on the expression of the proapoptotic protein Bax [29,30]. This, combined with a decrease in the levels of antiapoptotic proteins in the cell resulting from an NF-κB blockade, leads to an imbalance in favor of proapoptotic proteins, resulting in an increase in their activity, the formation of protein-permeable pores in the mitochondrial membrane, and the release of cytochrome c into the intracellular environment [23,26]. The present investigation was conducted to determine the percentage of MCF-7 and MDA-MB-231 breast cancer cells with an active form of NF-κB protein after 24 h of exposure to the tested compounds (EDA-71 and cisplatin) at concentrations of 1.5 and 3 µM. The results of our analysis indicate that only the compound EDA-71 significantly decreased the activity of the transcription factor NF-κB in both tested breast cancer cell lines (Figure 8). In MCF-7 cells, 95.0 ± 0.4% of cells with active NF-κB were detected in the control group, but after treatment with EDA-71 at concentrations of 1.5 and 3 µM, this percentage decreased to 87.4 ± 1.4% and 83.3 ± 4.0%, respectively. In the breast cancer cell line, MDA-MB-231, 93.2 ± 0.5% of cells with an active form of NF-κB were observed after the use of 1.5 µM EDA-71, while treatment with a higher dose (3 µM) resulted in a drastic decrease in the number of cells with this active protein to 52.9 ± 0.9% (control: 95.7 ± 0.2%). Regarding the effect of the reference compound, no significant decrease in NF-κB activation was observed after 24 h of incubation in comparison with the control group. The observed cell population with active NF-κB was 94.2 ± 1.3% (1.5 µM) and 93.6 ± 1.2% (3 µM), while it was 95.6 ± 1.1% (1.5 µM) and 94.5 ± 0.5% (3 µM) for MCF-7 and MDA-MB-231 breast cancer cells, respectively. The above results reveal that compound EDA-71 is an inhibitor of the transcription factor NF-κB, and its effect was more pronounced in MDA-MB-231 triple-negative breast cancer (Figure 8).

Cytochrome c is a hemoprotein localized on the outside of the inner mitochondrial membrane, acting as an electron transporter in the respiratory chain. Its binding at this site is possible due to its conjunction with cardiolipin, which has a negative charge. During apoptosis undergoing via the intrinsic pathway, this glycerolphospholipid is oxidized, resulting in a decrease in its affinity for cytochrome c and the release of this protein into the cell cytoplasm [26,31]. Following this incident, the binding of cytochrome c, Apaf-1 (Apoptotic protease activating factor 1), and procaspase 9 in the intracellular environment results in apoptosome formation, leading to the autocatalytic activation of caspase 9 [23,26]. The aim of the present experiment was the cytometric analysis of caspase 9 activity in MCF-7 and MDA-MB-231 breast cancer cells treated for 24 h with cisplatin and EDA-71 at concentrations of 1.5 and 3 µM. The results show that only compound EDA-71 induced a significant increase in caspase 9 activity in MCF-7 and MDA-MB-231 cells (Figure 9). EDA-71-mediated caspase 9 activation at a concentration of 1.5 µM was observed in 3.2 ± 0.6% (MCF-7) and 9.4 ± 0.9% (MDA-MB-231) of the cancer cell population, whereas a concentration of 3 µM of this tested compound resulted in an increase to 12.7 ± 2.2% and 17.2 ± 1.4% in MCF-7 and MDA-MB-231 cells, respectively. In contrast, in MCF-7 estrogen-dependent breast cancer cells, the percentage of cells with active caspase 9 was similar to the control (1.0 ± 0.1%) for cisplatin at each concentration. These values were 1.1 ± 0.1% and 1.5 ± 0.3% for concentrations of 1.5 and 3 µM, respectively. The same situation was also observed in MDA-MB-231 estrogen-independent breast cancer cells, where the percentage of the control group with active caspase 9 was 5.5 ± 1.0%, while it was 6.6 ± 1.1% (1.5 µM) and 7.0 ± 0.9% (3 µM) in cells treated with cisplatin. As indicated by the obtained results and Figure 9, the observed increase in caspase 9 activity in estrogen-dependent (MCF-7) and estrogen-independent (MDA-MB-231) breast cancer cells is consistent with the results found in the AV/PI and MMP change assays, giving reason to conclude that apoptosis induced by EDA-71 may follow an intrinsic pathway.

After activation of initiator caspases (e.g., caspases 8, 9, and 10), both apoptotic pathways (intrinsic and extrinsic) converge into a common one and the executive phase of apoptosis begins. During this stage, active executioner caspases—mainly caspase 3, but also caspases 6 and 7—are formed. Their activation subsequently results in the exposure of PS on the extracellular side of the cell membrane and the destruction of structural and enzymatic proteins of the cell, finally causing the complete disintegration of the cell [23,24]. In the present study, our team decided to investigate by flow cytometry the activity of the executioner caspases 3/7 in MCF-7 and MDA-MB-231 breast cancer cells after 24 h of treatment with cisplatin and EDA-71 at two concentrations (1.5 and 3 µM). At the beginning, it should be highlighted that MCF-7 breast cancer cells do not express caspase 3 [32,33], so the activity of the executioner caspases is associated with the induction of an active form of caspase 7. As presented in Figure 10, an increase in caspase 3/7 activity (in MCF-7 only caspase 7) was only observed in MCF-7 and MDA-MB-231 breast cancer cells treated with EDA-71. In MCF-7 cells at a concentration of 1.5 µM of the tested compound, caspase 7 activity increased 2-fold (4.1 ± 0.6%) compared with the control, while at the higher concentration the increase was already almost 8-fold (14.9 ± 1.6%), whereas, in MDA-MB-231 cells, caspase 3/7 activation reached 8.5 ± 1.1% (1.5 µM) and 16.1 ± 1.0% (3 µM) of the cell population. In contrast, cisplatin at both concentrations did not induce any changes in caspase 7 activity in MCF-7 cells, and the noted values were the same as in the control group (1.9 ± 0.2/0.3%). Furthermore, a non-significant increase in caspase 3/7 activity was observed in MDA-MB-231 cells compared with the control (6.4 ± 0.2%), which was 6.5 ± 0.4% and 7.2 ± 0.7% of the tested cell population for 1.5 and 3 µM concentrations, respectively. The above results correspond well with the other data obtained from the analysis of the apoptotic process and the molecules involved in it, indicating that the novel selenocompound EDA-71 induces apoptosis in MCF-7 and MDA-MB-231 breast cancer cells proceeding through two pathways, extrinsic and intrinsic.

### 3.3. EDA-71 Induces Autophagy in Breast Cancer Cells

The activity of a significant part of clinically used and newly designed anticancer drugs is mainly based on the induction of apoptosis. Unfortunately, cancer cells can develop resistance to this process by activating and enhancing prosurvival pathways. One way to overcome this phenomenon is to trigger other types of cell death. There is evidence in the available literature that excessive activation of the autophagy process in a cancer cell can lead to its “killing.” For this reason, induction of autophagy along with simultaneous apoptosis could improve the efficacy of anticancer treatment and/or overcome the chemoresistance of cancer cells [34,35]. As this process could be one of the components of the compound’s anticancer activity, we decided to determine by flow cytometry the state of autophagy of MCF-7 and MDA-MB-231 breast cancer cells treated for 24 h with EDA-71 and cisplatin at two concentrations (1.5 and 3 µM). Significant induction of autophagy was only observed in MCF-7 and MDA-MB-231 breast cancer cells treated with selenocompound EDA-71 (MCF-7: 3µM; MDA-MB-231: 1.5 and 3 µM; Figure 11). EDA-71 at a concentration of 1.5 µM showed no significant effect on the change of autophagy status in MCF-7 cells (1.6 ± 0.6% of autophagic cells), whereas, after treatment with the tested compound at a concentration of 3 µM, 4.5 ± 1.7% of cells with an active autophagy process were detected, which represented a 4-fold increase in this population compared with the control group (1.2 ± 0.1%). A higher percentage of cells that formed autophagosomes and autolysosomes as a result of EDA-71 activity was found in MDA-MB-231 triple-negative breast cancer cells, with values of 5.8 ± 1.2% (a 4-fold increase compared with control; control: 1.9 ± 0.5%) and 23.5 ± 4.0% (a 12-fold increase compared with control) for concentrations of 1.5 and 3 µM, respectively. For cisplatin treatment, there was no significant effect on autophagy induction in any of the tested cell lines at either of the two concentrations, and the percentages of autophagic cells were similar at the doses used (Figure 11). In MCF-7 cells, it was 2.0 ± 0.4% (1.5 µM) and 2.4 ± 0.2% (3 µM), while in MDA-MB-231, it was 2.6 ± 0.5% and 2.7 ± 0.7% for concentrations of 1.5 and 3 µM, respectively.

The mammalian target of rapamycin (mTOR) is one of the major proteins that regulate autophagy and metabolism in the cell. It has been found that many cancers exhibit up-regulation of the PI3K/Akt/mTOR signaling pathway, resulting in the inhibition of autophagy by mTOR protein in cancer cells and their survival [36]. Hence, to confirm the magnitude of autophagy induced by the tested compound, we decided to assess the extent of mTOR inhibition in MCF-7 and MDA-MB-231 breast cancer cells after treatment with EDA-71 and cisplatin (24 h; 1.5 and 3 µM). The obtained results correlate with the degree of autophagy induction observed in the previous study, as the number of autophagosomes and autolysosomes present in breast cancer cells increased (Figure 11) with the rise of mTOR inhibition (mTOR negative cells; Figure 12). After EDA-71 treatment, 4.6 ± 0.7% (1.5 µM) and 5.2 ± 1.5% (3 µM) of cells with inhibited mTOR were observed in MCF-7 cells. In turn, in MDA-MB-231 cells, the percentage of mTOR-negative cells was higher and was 9.0 ± 0.5% and 11.3 ± 0.2% for EDA-71 at 1.5 and 3 µM, respectively. For the exposure of both breast cancer cell lines to cisplatin, the received values confirmed the absence of significant induction of autophagy and were similar (MCF-7: 2.3 ± 0.3% and 3.3 ± 0.7%, MDA-MB-231: 3.8 ± 0.6% and 4.5 ± 0.6% of mTOR-negative cells for 1.5 and 3 µM concentrations, respectively) to the control group (MCF-7: 1.7 ± 0.3%, MDA-MB-231: 3.7 ± 1.2%). Concluding, the high cytotoxicity and efficacy of the novel compound EDA-71, especially against the chemoresistant MDA-MB-231 triple-negative breast cancer cells, may occur due to the simultaneous induction of apoptosis and autophagy processes.

### 3.4. EDA-71 Induces Cell Cycle Arrest in Breast Cancer Cells

The cell cycle is an ordered cyclic sequence of events leading to cell division, including cancer cells, which makes it a key process responsible for their proliferation and growth. Consistent with this, disrupting the cell cycle by arresting this process may be a strategy used for anticancer treatment [37]. To determine the exact molecular mechanism of anticancer activity of the novel selenium compound, the distribution of cell cycle phases in MCF-7 and MDA-MB-231 breast cancer cells after 24 h of exposure to EDA-71 and cisplatin (1.5 and 3 µM) was investigated by flow cytometry. The obtained results indicate that both tested compounds affected the cell cycle in a dose-dependent manner (Figure 13). Treatment of MCF-7 cells with 1.5 µM EDA-71 led to an insignificant increase in the cell population in the G_2_/M phase of the cell cycle from 22.9 ± 2.2% to 29.3 ± 1.4% compared with the control group, while use of this compound at a concentration of 3 µM resulted in its significant increase to 54.4 ± 0.7%. In MDA-MB-231 cells, it was also observed that the cell population in the S phase of the cell cycle increased insignificantly from 20.5 ± 0.1% to 22.8 ± 0.6% compared with the control after treatment with 1.5 µM of the tested selenoester, while the increase in its dose to 3 µM caused a significant increase in this percentage to 44.5 ± 1.8%. While EDA-71 led to an increase in the population of cells in the G_2_/M phase (MCF-7) or S phase (MDA-MB-231) of the cell cycle, a simultaneous decrease in the percentage of cells in the G_1_ and S phase (MCF-7 cells) or the G_1_ and G_2_/M phase (MDA-MB-231 cells) was observed. In contrast, when the tested cancer cells were exposed to cisplatin, we observed that it arrested them in the S phase of the cell cycle in both breast cancer cell lines (MCF-7: 50.4 ± 5.4% and 70.7 ± 3.9%; MDA-MB-231: 36.3 ± 0.3% and 46.3 ± 2.5% of the cell population in this phase of the cell cycle). To sum up, the effect of the novel compound EDA-71 was dependent on the tested breast cancer cell line and led to cell arrest in the G_2_/M and S phase of the cell cycle in MCF-7 and MDA-MB-231 cells, respectively. Moreover, the tested compound affected this process in a dose-dependent manner. All mean values of the investigated cell cycle distribution with their standard deviations are shown in Figure 13.

To confirm the effect of the new selenoester on cell arrest in different phases of the cell cycle, cyclin E1 and cyclin A2 activity was analyzed by flow cytometry in MCF-7 and MDA-MB-231 breast cancer cells (24 h; EDA-71 and cisplatin at 1.5 and 3 µM). The analyzed cyclins are regulatory proteins responsible for cell cycle progression as a result of the activation of the respective cyclin-dependent kinases (CDKs), and their highest activity is observed during the transition of the cell from the G_1_ to the S phase (cyclin E1) and in the G_2_ phase (cyclin A2) [38]. During the analysis of the obtained results, a similar decrease in cyclin E1 activity and a comparable increase in the active form of cyclin A2 following cisplatin and EDA-71 treatment of MDA-MB-231 cells were observed (Figure 14 and Figure 15), which confirms the arrest of these cells in the S phase of the cell cycle. In contrast, a much lower percentage of cyclin E1 positive cells and a distinctly higher percentage of cyclin A2 positive cells in MCF-7 breast cancer cells after incubation with EDA-71 compared with cisplatin were observed (Figure 14 and Figure 15), indicating that EDA-71 and cisplatin arrest the population of these cells in the G_2_/M and S phase, respectively.

## 4. Discussion

The steadily increasing incidence of cancer appears to be a worrying and global health problem. This is why numerous research teams are still taking steps to reduce this disease entity. One of the widely used therapeutic approaches in the treatment of cancer is chemotherapy, and the search for an effective anticancer drug is setting new directions of research in the field of oncology. An important position in the biological and biomedical processes of cancer is now occupied by Se compounds. Recent reports suggest an important role for Se in the chemoprevention of many cancers, including breast cancer. The mechanism would be based mainly on the maintenance of the oxidative–antioxidant balance in the body, the inhibition of angiogenesis, and the induction of the apoptotic process in cancer cells. Therefore, it is believed that compounds containing this element could be potential chemopreventive or anticancer agents [6].

Previously, it was shown that selenium compounds have been known to be highly cytotoxic toward a variety of cancer cell lines [12,19,40,41,42]. In this study, we demonstrated that both EDA-71 and E-NS-4 selenium compounds exhibited cytotoxic activity on breast cancer cell lines (MCF-7 and MDA-MB-231), with IC_50_ values in terms of low micromolar concentrations (1.4 to 4.52 µM). We utilized a cell line that was more sensitive to the compound MCF-7 and one that was most resistant to MDA-MB-231 in order to determine if the compounds elicited different responses that may display their resistance or sensitivity to the compounds. Unfortunately, the study showed that the tested compounds did not show selectivity to the analyzed breast cancer cells. Nevertheless, it is significant that they show high cytotoxic activity at lower concentrations than cisplatin. Moreover, tested compounds EDA-71 and E-NS-4 exhibited comparable cytotoxicity in both normal breast epithelial and breast cancer cells. Similar observations were reported by the team of Manda et al. [43] who studied the effects of another selenocompound, sodium selenite, in A549 lung cancer cell line and BEAS-2B normal human bronchial epithelial cells. However, the broad toxicity of these compounds may be overcome if the novel selenoesters were to be used in future clinical trials. Shortcomings frequently encountered with anticancer compounds, such as normal tissue toxicity, as well as the high incidence rate of drug resistance, may be overcome through the use of lipid-based nanocarriers or other nanomaterials. Importantly, these systems are biocompatible and based on physiological and morphological differences in cancer cells, and the additional functionalization of their surface enables targeting against these cells, sparing normal cells [44,45]. In a study conducted by Tang et al. in 2022 [46], it was observed that the use of estrone-conjugated PEGylated liposome containing carboplatin increased the efficacy of this anticancer drug against ovarian cancer xenografts and significantly reduced acute toxicity in mice.

High cytotoxic activity can lead to cell death by several significantly different death pathways—apoptosis, entosis, necrosis, necroptosis (regulated necrosis), and autophagy—and the type of cell death depends on the properties of the chemical compound used or the dose applied [12,16,40,42]. The study by Shigemi et al. [47] showed that Se compounds direct cancer cells mainly to the apoptosis pathway. The studies of our team seem to confirm this thesis, as they demonstrated a high capacity of EDA-71 to induce apoptosis in the concentration range oscillating around 1.5–3.0 µM, with a simultaneous negligible ability to activate necrosis. Due to the fact that EDA-71 showed much higher cytotoxic activity in comparison with E-NS-4, we focused our further attention only on this compound. The results obtained by flow cytometry after 24 h of incubation with the tested compound showed a correlation with the results obtained by MTT. The results obtained with both methods indicate that the type of cell death shifts towards apoptosis as the concentration of the tested compound increases. This is probably related to the fact that EDA-71 leads to very severe cell dysfunction and the activation of proteins involved in the process of apoptosis or autophagy. Additionally, the high proapoptotic activity of EDA-71 was also present in normal human breast epithelial cells (MCF-10A). Despite this unfavorable fact, it should be noted that the selenium structure investigated in this study may provide an excellent molecular framework for a novel group of anticancer drugs in the future. Compounds with high proapoptotic activity are still considered an excellent basis for future anticancer drugs [48] and the high toxicity of compound EDA-71 can be reduced by using, for example, nanocarriers [45] as we discussed above.

The process of Se-activated apoptosis can follow an intrinsic (mitochondrial) and extrinsic (receptor) pathway [12,40,42,47]. In the intrinsic pathway of apoptosis, a decrease in mitochondrial membrane potential, release of cytochrome c, and an increase in caspase 9 and caspase 3 activity can be observed [23,32,49]. The mechanism of action of numerous selenoorganic compounds is related to the activation of the mitochondrial apoptotic pathway [42,47]. The study of Se derivative—ebselen—performed by Zhang et al. [50] revealed a decrease in mitochondrial membrane potential, release of cytochrome c, and an increase in caspase 9 activity for human multiple myeloma U266 and RPMI8226 cells. To assess the ability of compound EDA-71 to direct cells to the apoptotic pathway associated with the activation of the intrinsic pathway, mitochondrial membrane potential change assay and a measurement of caspase 9 and caspase 3 activities were performed. Loss of mitochondrial transmembrane potential (ΔΨ_m_) is considered an early marker of apoptosis [51]. Sakallı Çetin et al. [52] described a loss of ΔΨ_m_ that occurred in MCF-7 breast cancer cells treated with sodium selenite, while Zhang et al. [53] demonstrated a decrease in ΔΨ_m_ in human epithelial cervical cancer HeLa cells treated with Se-containing phenylindolyl ketone derivative. The results we obtained for compound EDA-71 showed that the mitochondrial membrane potential decreased in a dose-dependent manner. The consequent loss of ΔΨ_m_ was followed by the activation of caspases, including caspase 9 and then caspase 3, so the study of their activities was a further step to elucidate the mechanism of cell death. Our study of caspase 9 activity showed a significant increase in the activity of this protein. Along with its increase, a correlated increase in caspase 3 occurred, which is clear evidence that EDA-71 exhibits proapoptotic properties through the intrinsic pathway.

Caspase 3 is also activated by the action of caspases 8 and 10 in the extrinsic pathway of apoptosis. The activation of these caspases results from the dimerization of the FADD molecule and its interaction with the non-active forms of these proteins (procaspases 8 and 10). This leads to the formation of the DISC complex, which autocatalyzes and activates caspases 8 and 10 [23,24]. As a result of our research, we have shown that EDA-71 enables the expression of active caspases 8 and 10 in MCF-7 and MDA-MB 231 breast cancer cells. This may suggest that EDA-71 also induces programmed cell death by the caspase-dependent pathway of caspases 8 and caspase 10 (extrinsic pathway). In addition, these observations can be combined with the result of our study illustrating that EDA-71 affects the increase in the expression of active caspase 3.

Therapy with selenium compounds can affect the activity and levels of numerous proteins, including those important in apoptosis and autophagy [12,42,47]. A study conducted in 2015 by Wu et al. [54] showed that mTOR protein levels may be downregulated during Se-allylselenocysteine therapy, possibly due to AMPK protein inhibition. Thus, the authors of this article suggest that the decrease in mTOR protein levels can lead to autophagy resulting in cell death. Similar results were obtained by Lee and Facompre with their co-workers, but they also demonstrated an additional role in the inhibition of the PI3K/Akt/mTOR signaling pathway by Se-containing compounds [55,56]. The results obtained in our work indicate that EDA-71 can lead to a reduction in mTOR protein levels, a factor that may promote cell death by autophagy. Knowledge of the involvement of this process in the elimination of EDA-71-treated cells seems to be of utmost importance, since modulation of signal transduction pathways involved in autophagy, on the one hand, enables the additional sensitization of cells to EDA-71 and, on the other hand, may facilitate the elimination of the resistance phenomenon. Interestingly, while the autophagy process was already observed at a concentration of 1.5 μM in the case of MDA-MB-231 cells, this process was activated in the case of MCF-7 cells over a concentration of 3.0 μM. The obtained results suggest that one of the possible cell death pathways under the influence of EDA-71 may be the process of autophagy, which significantly depends on the dose used. 

Considering the results obtained in MTT and apoptosis assays, we decided to determine the effect of compound EDA-71 on the cell cycle. A study of the effects of EDA-71 on cell cycle phases showed cell cycle inhibition in the G_2_/M phase in MCF-7 cells and the S phase for MDA-MB-231 cells. Similar results were obtained with selenocystine treatment of the same cell line [57]. The results obtained for the compound EDA-71 suggest that it affects specific phases of the cell cycle. Analysis of cell cycle phases showed an increase in the number of cells stained with propidium iodide with increasing dose.

The studies conducted so far in this work suggest that EDA-71 is a compound with very interesting anticancer activity. Due to the fact that it is a completely new compound with biological activity that is difficult to predict, the experiments conducted within the framework of this work covered many aspects related to its activity (the potential molecular mechanism of its anticancer activity is summarized in Figure 16), so that it is possible to delineate the most important directions in which further research should be directed. First of all, an unfavorable fact in the potential future clinical application of this agent is that it has high toxicity to normal cells, so as a first step it would be appropriate to undertake to reduce this undesirable phenomenon. The development of a targeted drug delivery system based, for example, on lipid nanocarriers, could help overcome the cytotoxicity of this compound against normal cells with simultaneous preservation of high anticancer activity. In the next stage, *in vitro* studies evaluating the developed formulation of the drug and the associated reduction of toxic side effects should be planned, and if the results are satisfactory, *in vivo* studies should be conducted. In addition, further detailed studies should be carried out to elucidate the mechanism of cytotoxic activity in cancer cells, as well as to allow for evaluation of the application of the tested compound with the developed drug formulation as an effective anticancer drug candidate.

## 5. Conclusions

In the present study, we aimed to evaluate the cytotoxic activity and antiproliferation effects of two selenoesters differing in the end fragment against MCF-7 and MDA-MB-231 breast cancer cells. The compound containing the ketone end fragment (EDA-71) was more cytotoxic and exhibited a stronger inhibitory effect on the cell proliferation of both breast cancer lines than the selenoester containing the nitrile end fragment (E-NS-4), but, importantly, both were very active at low micromolar concentrations (<5 µM). Similar results were observed in normal human breast epithelial cells (MCF-10A). In further experiments performed with the more active compound (EDA-71), a strong induction of apoptosis was observed, especially in MDA-MB-231 triple-negative breast cancer cells and normal MCF-10A cells. This process follows both an extrinsic pathway with activation of caspases 8 and 10 and an intrinsic pathway, as evidenced by a decrease in transcription factor NF-κB activity and mitochondrial potential and an increase in caspase 9 activity. Additionally, increased activity of caspases 3 and 7, which are common executioner caspases for both apoptotic pathways, was observed. 

Furthermore, another investigation of the molecular mechanism of action of EDA-71 revealed that its autophagy activity is dose-dependent, and one possible target is the inhibition of mTOR protein. Apart from the above, our study also showed that the novel selenoorganic compound—EDA-71—can exhibit its anticancer activity through cell cycle arrest, and its effect on specific phases of the cell cycle was dependent on the type of breast cancer line (MCF-7, G_2_/M phase; MDA-MB-231, S phase). Confirmation of cell arrest in specific phases of the cell cycle was observed by a decrease in cyclin E1 levels with a concomitant increase in cyclin A2 levels.

To conclude, the tested novel selenoester EDA-71 is a highly interesting compound whose probably broad, molecular mechanism of action has not yet been fully understood, prompting further research in this field. The above results, as well as the very low doses of the agent used, suggest that EDA-71 seems to be a really promising candidate as a future potential anticancer drug for breast cancer therapy. However, this requires extensive research involving this substance.

## Figures and Tables

**Figure 1 cancers-14-04304-f001:**
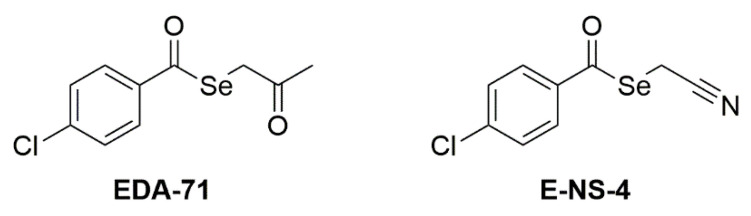
Structure of the tested compounds.

**Figure 2 cancers-14-04304-f002:**
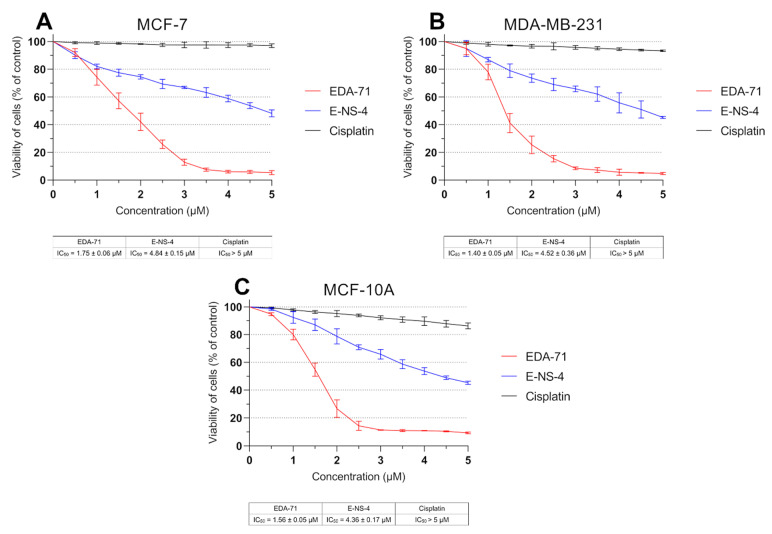
Viability of MCF-7 (**A**) and MDA-MB-231 (**B**) breast cancer cells and MCF-10A normal breast epithelial cells (**C**) incubated for 24 h with the tested compounds and cisplatin at various concentrations. Results are presented as mean values (percent of untreated cells) ± SD obtained from three independent experiments (*n* = 3) conducted in triplicate.

**Figure 3 cancers-14-04304-f003:**
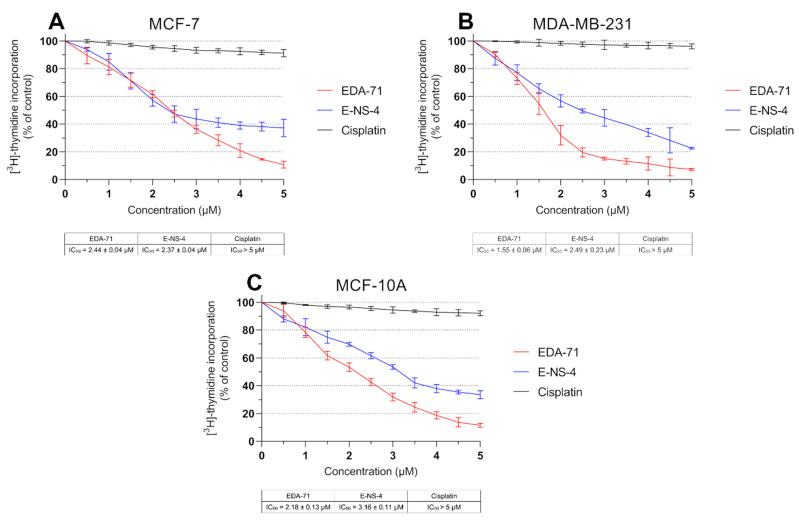
DNA biosynthesis process (incorporation of [^3^H]-thymidine into DNA) observed in MCF-7 (**A**) and MDA-MB-231 (**B**) breast cancer cells and MCF-10A normal breast epithelial cells (**C**) incubated for 24 h with the tested compounds and cisplatin at various concentrations. Results are presented as mean values (percent of untreated cells) ± SD obtained from three independent experiments (*n* = 3) conducted in triplicate.

**Figure 4 cancers-14-04304-f004:**
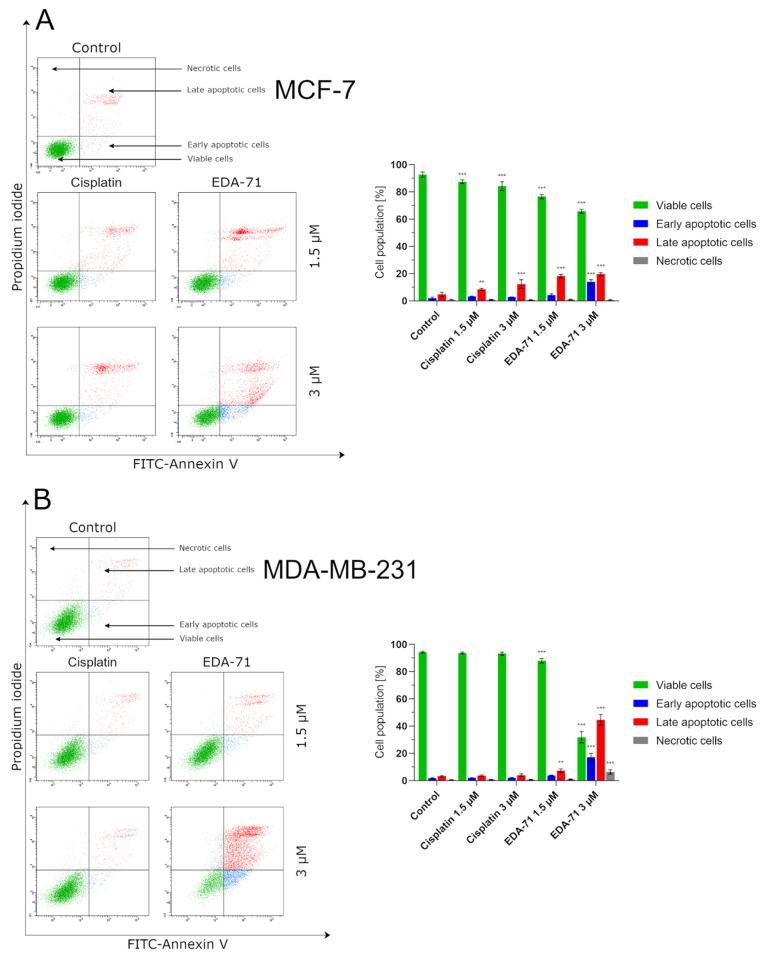
Induction of apoptosis in MCF-7 (**A**) and MDA-MB-231 (**B**) breast cancer cells and MCF-10A normal breast epithelial cells (**C**) incubated for 24 h with EDA-71 and cisplatin (1.5 and 3 µM). The experiment was performed using Annexin V-FITC/propidium iodide double staining and a flow cytometer. Results are presented as mean values ± SD obtained from three independent experiments (*n* = 3) conducted in triplicate. Statistical differences between the experimental (treated cells) and control (untreated cells) groups were determined using one-way ANOVA followed by Dunnett’s test. ** *p* < 0.01 vs. control group, *** *p* < 0.001 vs. control group.

**Figure 5 cancers-14-04304-f005:**
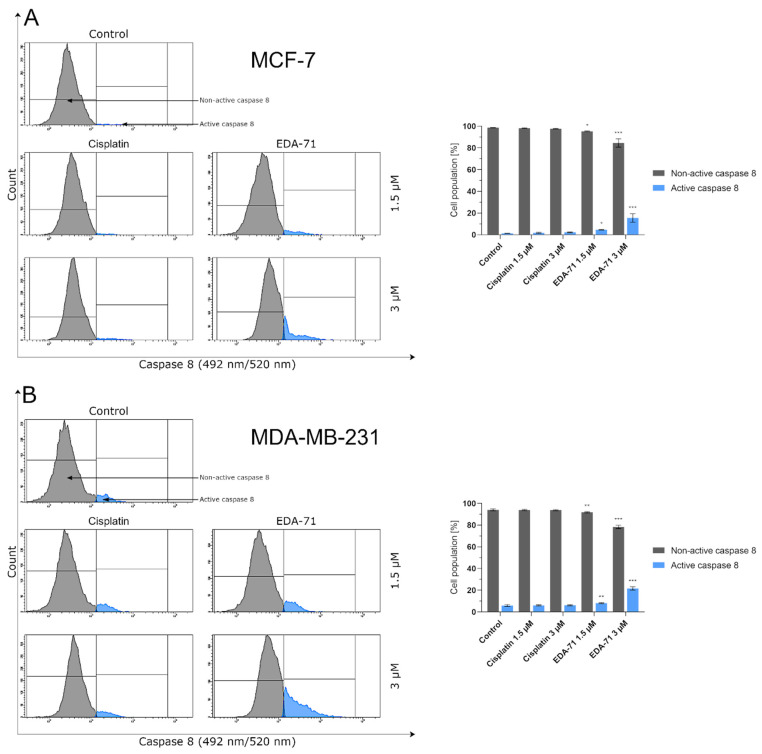
Cytometric analysis of caspase 8 activity in MCF-7 (**A**) and MDA-MB-231 (**B**) breast cancer cells after 24 h incubation with EDA-71 and cisplatin (1.5 and 3 μM). Results are presented as mean values ± SD obtained from three independent experiments (*n* = 3) conducted in triplicate. Statistical differences between the experimental (treated cells) and control (untreated cells) groups were determined using one-way ANOVA followed by Dunnett’s test. * *p* < 0.05 vs. control group, ** *p* < 0.01 vs. control group, *** *p* < 0.001 vs. control group.

**Figure 6 cancers-14-04304-f006:**
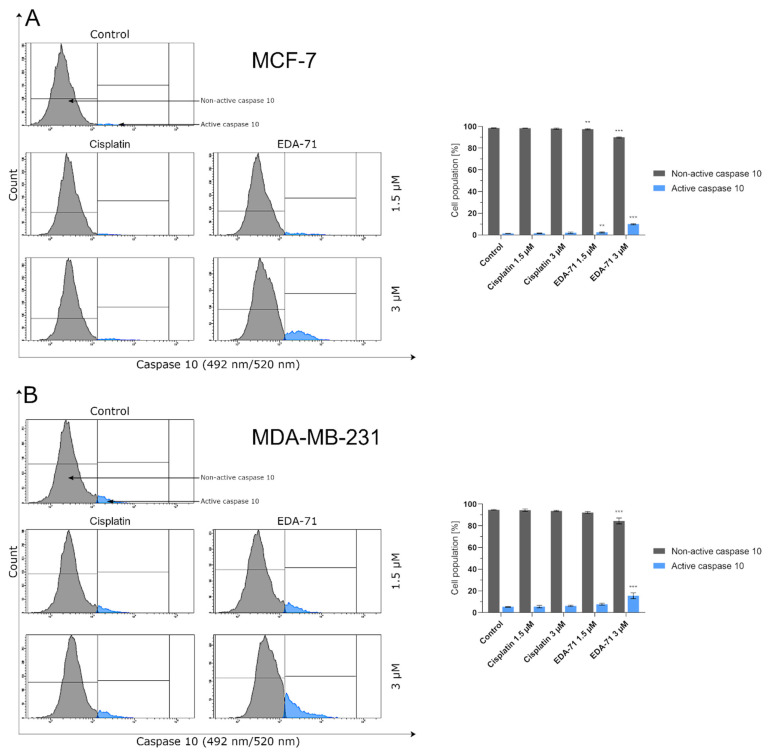
Cytometric analysis of caspase 10 activity in MCF-7 (**A**) and MDA-MB-231 (**B**) breast cancer cells after 24 h incubation with EDA-71 and cisplatin (1.5 and 3 μM). Results are presented as mean values ± SD obtained from three independent experiments (*n* = 3) conducted in triplicate. Statistical differences between the experimental (treated cells) and control (untreated cells) groups were determined using one-way ANOVA followed by Dunnett’s test. ** *p* < 0.01 vs. control group, *** *p* < 0.001 vs. control group.

**Figure 7 cancers-14-04304-f007:**
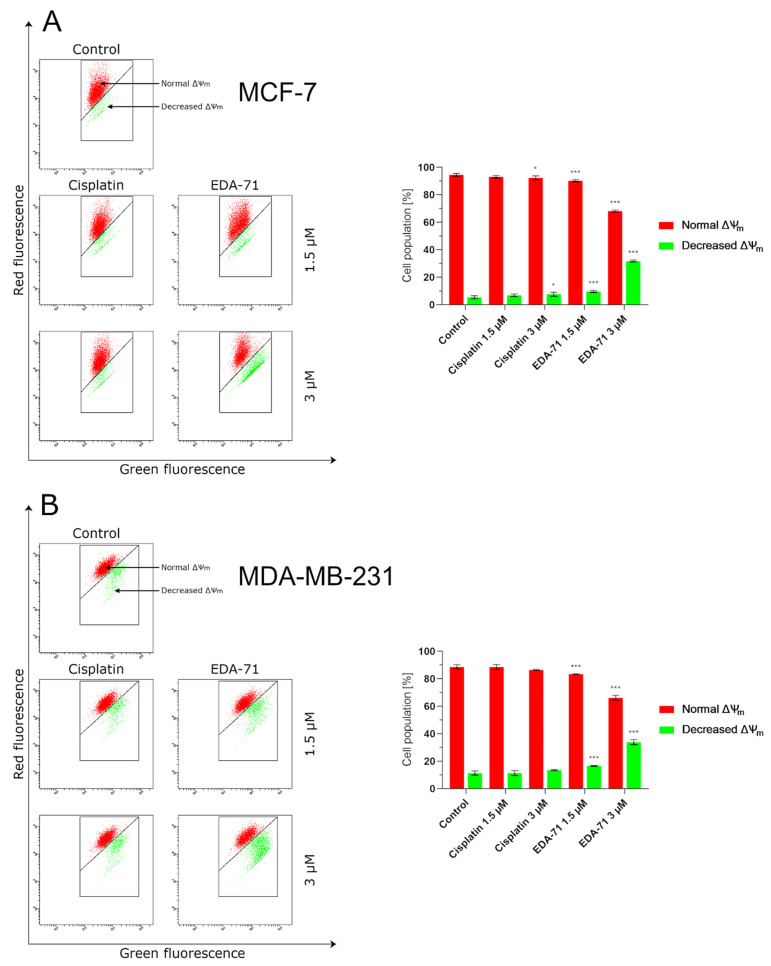
Cytometric analysis of mitochondrial membrane potential (MMP, ∆Ψ_m_) changes in MCF-7 (**A**) and MDA-MB-231 (**B**) breast cancer cells after 24 h incubation with EDA-71 and cisplatin (1.5 and 3 μM). Results are presented as mean values ± SD obtained from three independent experiments (*n* = 3) conducted in triplicate. Statistical differences between the experimental (treated cells) and control (untreated cells) groups were determined using one-way ANOVA followed by Dunnett’s test. * *p* < 0.05 vs. control group, *** *p* < 0.001 vs. control group.

**Figure 8 cancers-14-04304-f008:**
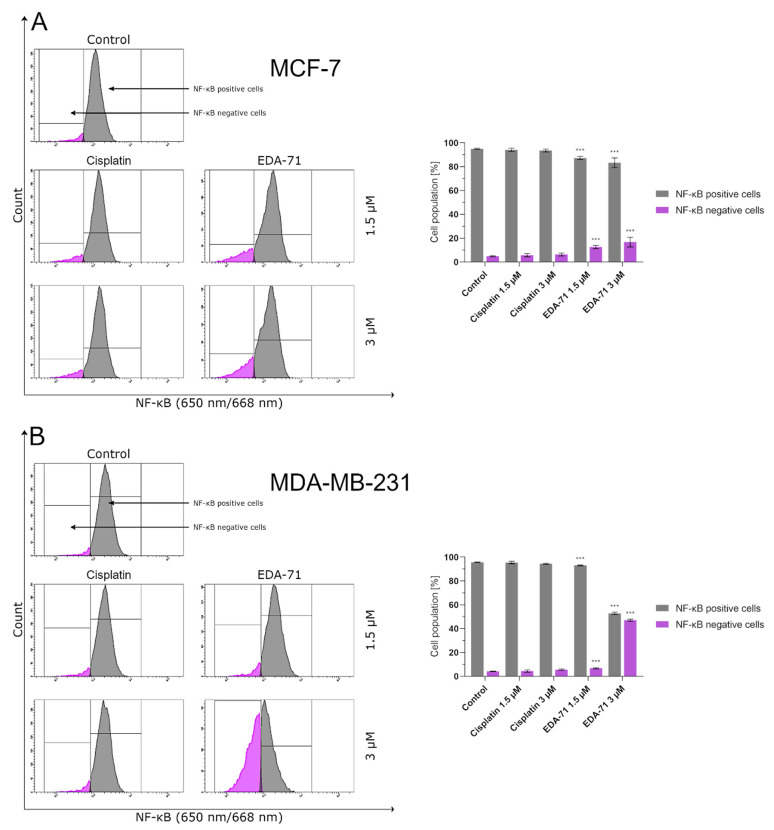
Cytometric analysis of NF-κB activity in MCF-7 (**A**) and MDA-MB-231 (**B**) breast cancer cells after 24 h incubation with EDA-71 and cisplatin (1.5 and 3 μM). Results are presented as mean values ± SD obtained from three independent experiments (*n* = 3) conducted in triplicate. Statistical differences between the experimental (treated cells) and control (untreated cells) groups were determined using one-way ANOVA followed by Dunnett’s test. *** *p* < 0.001 vs. control group.

**Figure 9 cancers-14-04304-f009:**
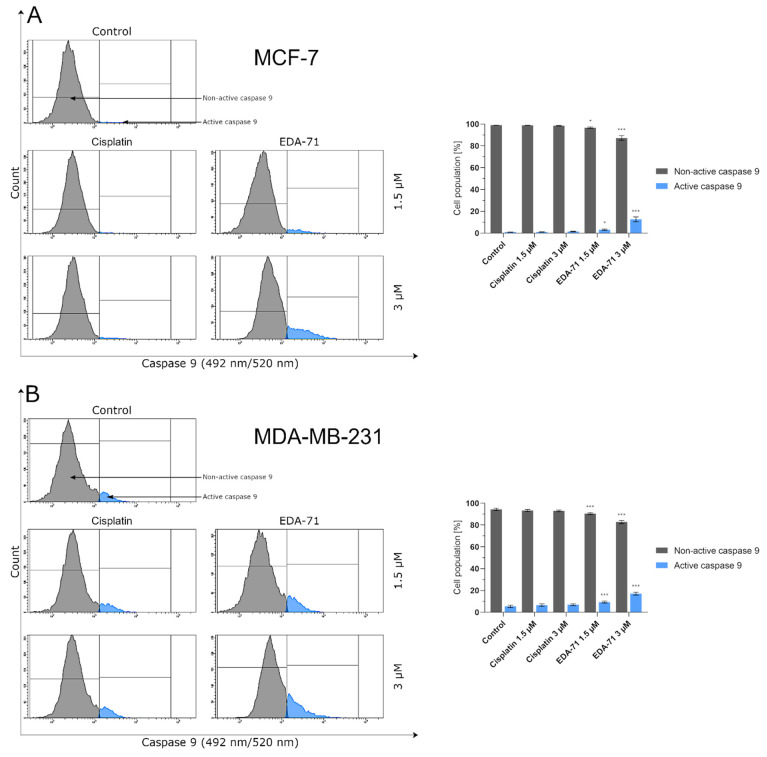
Cytometric analysis of caspase 9 activity in MCF-7 (**A**) and MDA-MB-231 (**B**) breast cancer cells after 24 h incubation with EDA-71 and cisplatin (1.5 and 3 μM). Results are presented as mean values ± SD obtained from three independent experiments (*n* = 3) conducted in triplicate. Statistical differences between the experimental (treated cells) and control (untreated cells) groups were determined using one-way ANOVA followed by Dunnett’s test. * *p* < 0.05 vs. control group, *** *p* < 0.001 vs. control group.

**Figure 10 cancers-14-04304-f010:**
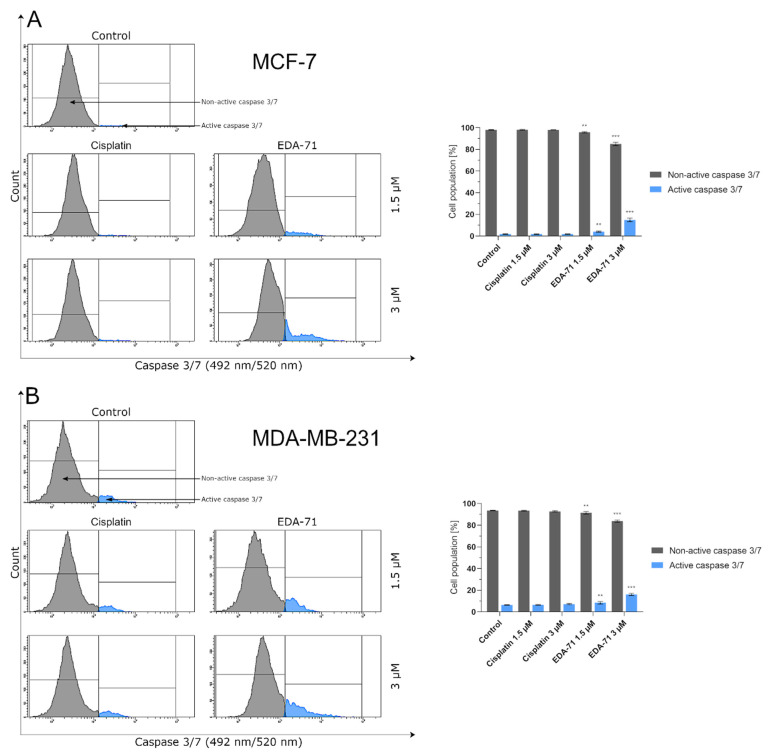
Cytometric analysis of caspase 3/7 activity in MCF-7 (**A**) and MDA-MB-231 (**B**) breast cancer cells after 24 h incubation with EDA-71 and cisplatin (1.5 and 3 μM). Results are presented as mean values ± SD obtained from three independent experiments (*n* = 3) conducted in triplicate. Statistical differences between the experimental (treated cells) and control (untreated cells) groups were determined using one-way ANOVA followed by Dunnett’s test. ** *p* < 0.01 vs. control group, *** *p* < 0.001 vs. control group.

**Figure 11 cancers-14-04304-f011:**
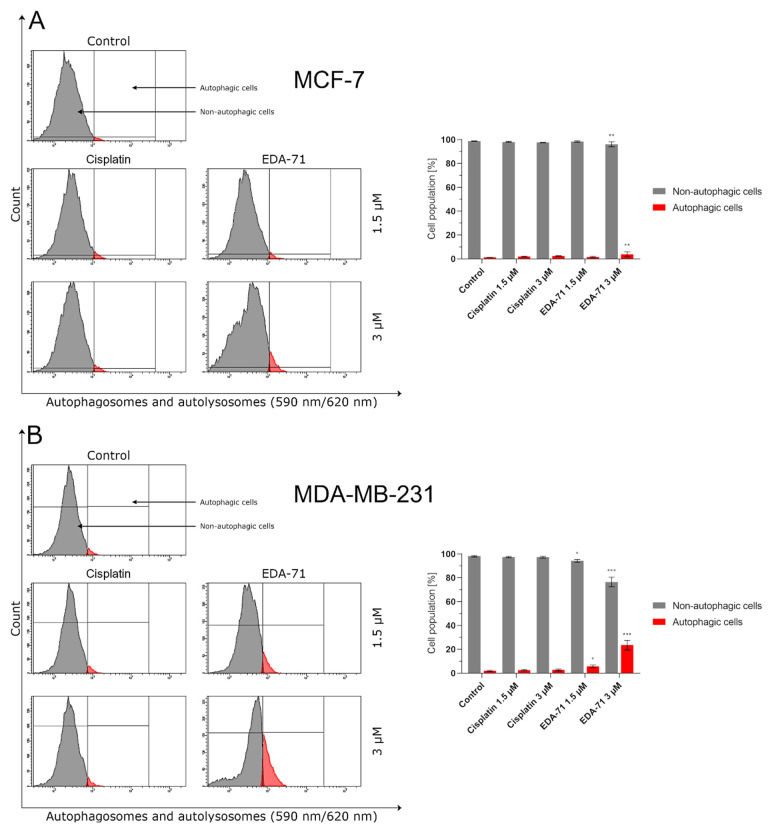
Induction of autophagy in MCF-7 (**A**) and MDA-MB-231 (**B**) breast cancer cells incubated for 24 h with EDA-71 and cisplatin (1.5 and 3 µM). The experiment was performed using Autophagy Probe staining and a flow cytometer. Results are presented as mean values ± SD obtained from three independent experiments (*n* = 3) conducted in triplicate. Statistical differences between the experimental (treated cells) and control (untreated cells) groups were determined using one-way ANOVA followed by Dunnett’s test. * *p* < 0.05 vs. control group, ** *p* < 0.01 vs. control group, *** *p* < 0.001 vs. control group.

**Figure 12 cancers-14-04304-f012:**
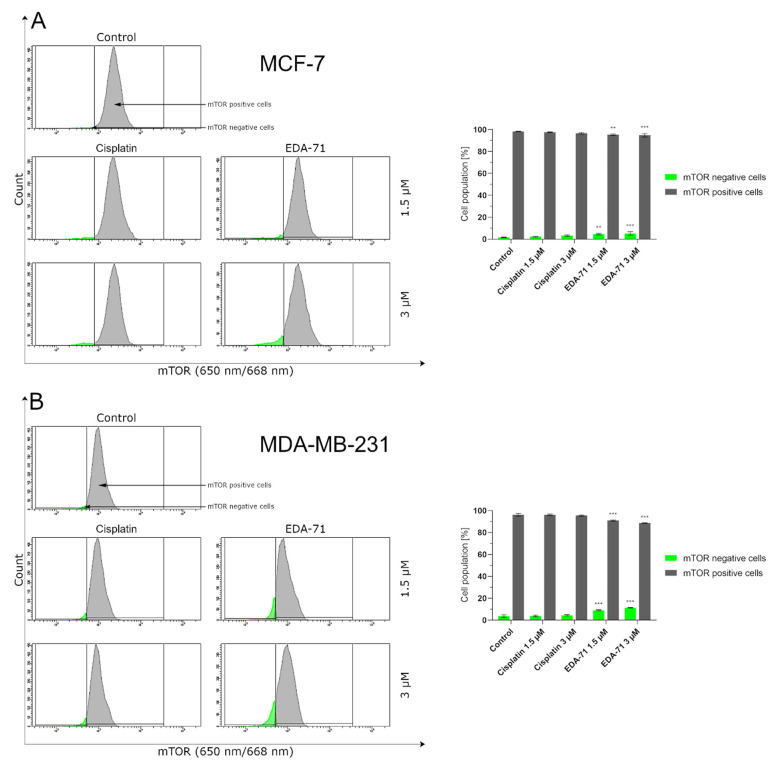
Cytometric analysis of mTOR activity in MCF-7 (**A**) and MDA-MB-231 (**B**) breast cancer cells after 24 h incubation with EDA-71 and cisplatin (1.5 and 3 μM). Results are presented as mean values ± SD obtained from three independent experiments (*n* = 3) conducted in triplicate. Statistical differences between the experimental (treated cells) and control (untreated cells) groups were determined using one-way ANOVA followed by Dunnett’s test. ** *p* < 0.01 vs. control group, *** *p* < 0.001 vs. control group.

**Figure 13 cancers-14-04304-f013:**
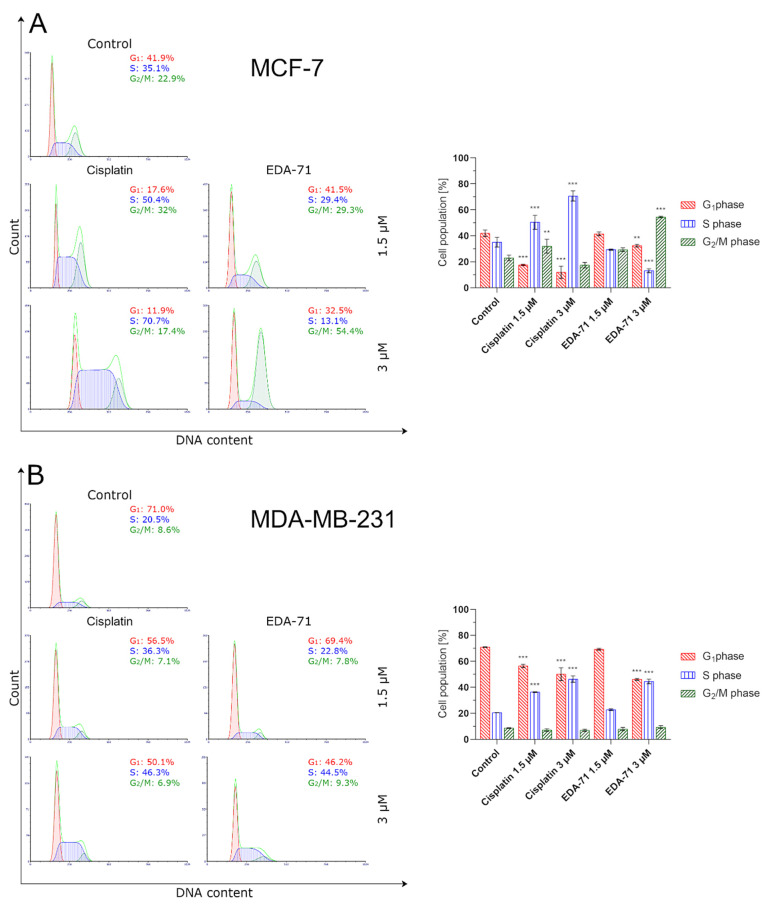
Cytometric analysis of cell cycle distribution in MCF-7 (**A**) and MDA-MB-231 (**B**) breast cancer cells after 24 h incubation with EDA-71 and cisplatin (1.5 and 3 μM). Results are presented as mean values ± SD obtained from three independent experiments (*n* = 3) conducted in triplicate. Statistical differences between the experimental (treated cells) and control (untreated cells) groups were determined using one-way ANOVA followed by Dunnett’s test. ** *p* < 0.01 vs. control group, *** *p* < 0.001 vs. control group.

**Figure 14 cancers-14-04304-f014:**
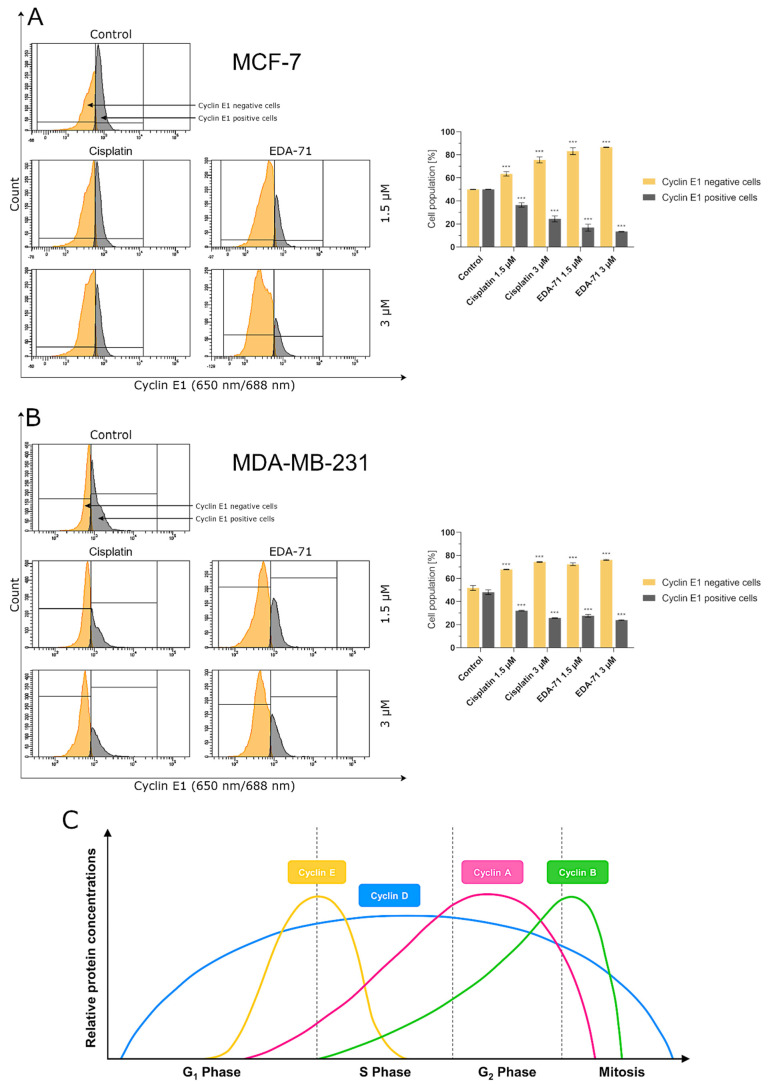
Cytometric analysis of cyclin E1 activity in MCF-7 (**A**) and MDA-MB-231 (**B**) breast cancer cells after 24 h incubation with EDA-71 and cisplatin (1.5 and 3 μM). Results are presented as mean values ± SD obtained from three independent experiments (*n* = 3) conducted in triplicate. Statistical differences between the experimental (treated cells) and control (untreated cells) groups were determined using one-way ANOVA followed by Dunnett’s test. *** *p* < 0.001 vs. control group. (**C**) illustrates changes in the relative concentrations of individual cyclins depending on the phase of the cell cycle (modified from [39]).

**Figure 15 cancers-14-04304-f015:**
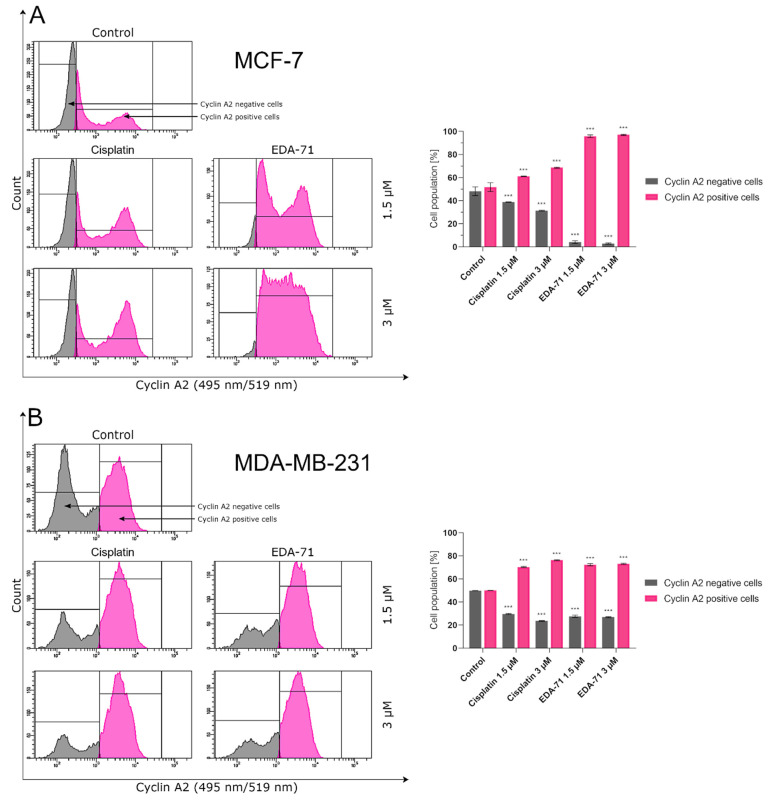
Cytometric analysis of cyclin A2 activity in MCF-7 (**A**) and MDA-MB-231 (**B**) breast cancer cells after 24 h incubation with EDA-71 and cisplatin (1.5 and 3 μM). Results are presented as mean values ± SD obtained from three independent experiments (*n* = 3) conducted in triplicate. Statistical differences between the experimental (treated cells) and control (untreated cells) groups were determined using one-way ANOVA followed by Dunnett’s test. *** *p* < 0.001 vs. control group.

**Figure 16 cancers-14-04304-f016:**
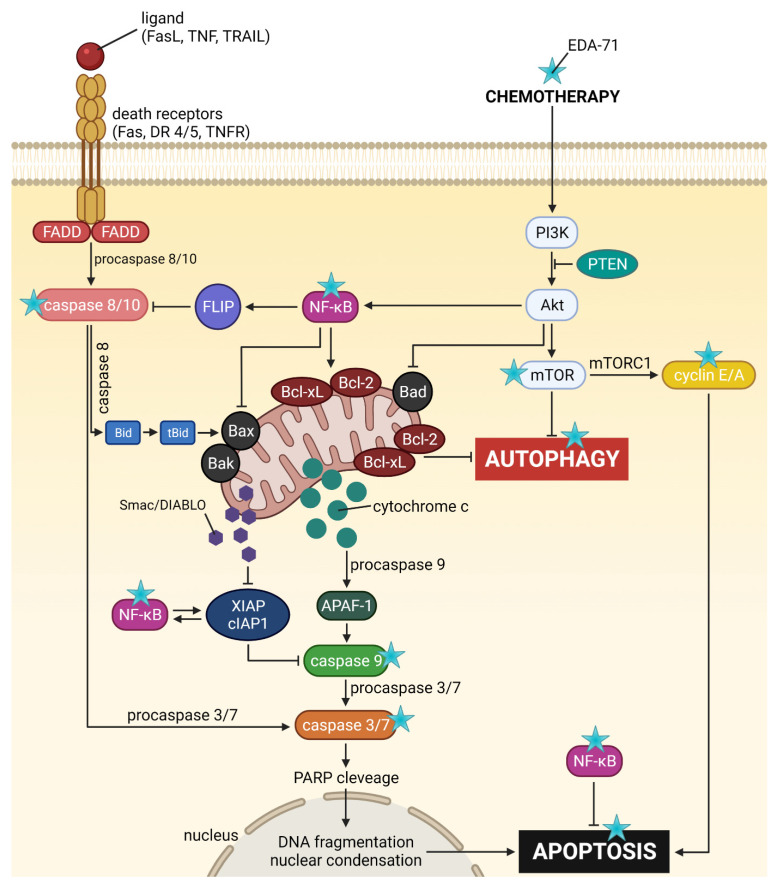
General scheme of the potential molecular anticancer activity of the novel selenoorganic derivative EDA-71.

## Data Availability

The data presented in this study are available in this article.

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
