# Peer review of "Novel Selenoesters as a Potential Tool in Triple-Negative Breast Cancer Treatment"

_cancers, 2022, doi:10.3390/cancers14174304_

Round 1
Reviewer 1 Report
This paper is an interesting mechanistic study into two novel selenoester compounds' mechanism of cytotoxicity in breast cancer cell lines. In summary, the two compounds seem to activate breast cancer cell death via intrinsic & extrinsic apoptosis, and somewhat through autophagy.
I do have a number of revisions, which are stated below:
Figure 3-15 - Should really all include normal cells (at least MCF10A). Figure 2 showed that the two novel selenoesters are still highly toxic to normal cells, and so all subsequent experiments should show the same. This is obviously a major revision. Thus, for this to be published in its current state, there must be a change of the text in body of paper to reflect this, suggestions on this are below.
Page 3, line 98 - self citation (12) of the importance of selenoesters is inappropriate. Please delete this sentence or re-state in a way that is not leading/suggestive.
Methods: How was statistical analysis done (ANOVA, T test, etc)? Which groups/subgroups were compared in the figures? This info should be included in both its own subsection within methods, and within each Figure legend.
Figures 5,6,9,10 - check figures for typo on “Caspase” (spelled capase in the figures).
Page 31, line 900 - this is overstated, as Figure 2 clearly showed significant chemotoxicity to normal cells. Please amend text.
Page 31, line 902 - a line must be added that states explicitly, without ambiguity, that the tested compounds show significant toxicity towards normal cells as well (Figure 2)
Discussion in general: As stated above, a major limitation of the study is that only one of the assays, calculating the ID50, was done on normal cells. The clinical utility of these drugs is essentially impossible to predict based on the following data. While it is an interesting proposition that selenoesters could be used as chemotherapy, their use would be incredibly hindered clinically if they cause the exact same toxicity to all normal cells. For this reason, the results cannot be overstated in the discussion, and this has to be directly addressed.
Page 32, line 997 - the data does not suggest this, this is speculation.
Page 33, line 1009 - the limitations, namely broad cytotoxicity to normal cells as well as shown in Figure 2, should be restated in this paragraph. Future directions, such as a way to overcome broad cytotoxicity (i.e. targeted drug delivery systems, localized administration, and further in vivo work) should be also discussed somewhere in the discussion.
Page 34, line 1029: must include toxicity to normal cells as shown in Figure 2.
Author Response
RESPONSE TO REVIEWER #1:
Thank you for your review of our manuscript entitled “Novel Selenoesters as a Potential Tool in Triple-Negative Breast Cancer Treatment” for publication in Cancers. Overall, we find Reviewer suggestions to be helpful, and constructive, and the corresponding revisions have strengthened the paper in multiple ways. According to your suggestions, we have expanded our manuscript with the content you suggested (the changes made in the manuscript are marked in red).
- “Figure 3-15 - Should really all include normal cells (at least MCF10A). Figure 2 showed that the two novel selenoesters are still highly toxic to normal cells, and so all subsequent experiments should show the same. This is obviously a major revision. Thus, for this to be published in its current state, there must be a change of the text in body of paper to reflect this, suggestions on this are below.”
Response: We agree with the Reviewer's opinion on this issue. Of course, it would be a major revision for us to perform all assays using normal cells, but we decided to investigate the incorporation of [3H]-thymidine into the DNA of these cells and assess the induction of the apoptosis process. The first analysis enabled us to determine another IC50 values of the tested compounds (in addition to the MTT assay), which allowed us to assess the toxicity of the tested compounds more precisely. In turn, the evaluation of the proapoptotic properties of the compounds by the Annexin V/propidium iodide assay is the basic test to which our further analyses refer, which allows us to conclude that the results of subsequent experiments should probably reveal the same. We hope that these two investigations will satisfy the Reviewer.
- “Page 3, line 98 - self citation (12) of the importance of selenoesters is inappropriate. Please delete this sentence or re-state in a way that is not leading/suggestive.”
Response: It has been corrected.
- “Methods: How was statistical analysis done (ANOVA, T test, etc)? Which groups/subgroups were compared in the figures? This info should be included in both its own subsection within methods, and within each Figure legend.”
Response: A subsection "2.13 Statistical Analysis" has been added to the section "2. Materials and Methods" and a description regarding statistical analysis has been included in the legend of each figure.
- “Figures 5,6,9,10 - check figures for typo on “Caspase” (spelled capase in the figures).”
Response: It has been corrected.
- “Page 31, line 900 - this is overstated, as Figure 2 clearly showed significant chemotoxicity to normal cells. Please amend text.”
Response: The sentence has been removed. A discussion of chemotoxicity to normal cells is described later in the discussion.
- “Page 31, line 902 - a line must be added that states explicitly, without ambiguity, that the tested compounds show significant toxicity towards normal cells as well (Figure 2).”
Response: It has been added.
- “Discussion in general: As stated above, a major limitation of the study is that only one of the assays, calculating the ID50, was done on normal cells. The clinical utility of these drugs is essentially impossible to predict based on the following data. While it is an interesting proposition that selenoesters could be used as chemotherapy, their use would be incredibly hindered clinically if they cause the exact same toxicity to all normal cells. For this reason, the results cannot be overstated in the discussion, and this has to be directly addressed.”
Response: We agree with the Reviewer's opinion on this issue, so we performed a study of [3H]-thymidine incorporation into the DNA of normal cells, which provided another IC50 values. They show that the toxicity to normal and cancer cells is comparable, so we described this and proposed possibilities to overcome this phenomenon in the discussion.
- “Page 32, line 997 - the data does not suggest this, this is speculation.”
Response: To avoid misleading the reader, the sentence has been removed.
- “Page 33, line 1009 - the limitations, namely broad cytotoxicity to normal cells as well as shown in Figure 2, should be restated in this paragraph. Future directions, such as a way to overcome broad cytotoxicity (i.e. targeted drug delivery systems, localized administration, and further in vivo work) should be also discussed somewhere in the discussion.”
Response: Discussion of limitations and further perspectives have been added.
- “Page 34, line 1029: must include toxicity to normal cells as shown in Figure 2.”
Response: It has been added.
We hope that our answer will satisfy Reviewer and Editor. We thank you for consideration and await your decision.

Reviewer 2 Report
This paper reports very interesting results with a new Selenium containing compound that seems to stimulate at very low concentrations also the apoptosis and autophagy of the MDA-MB-231 triple negative breast cancer cell lines.
It is great to see that this new compound works so well but of course there is no toxicity study and we can't know how it works in non-cancerogenic cells.
The authors mention the necessity of further studies. My suggestion is to emphasize more the possibility of high toxicity.
Author Response
RESPONSE TO REVIEWER #2:
Thank you for your review of our manuscript entitled “Novel Selenoesters as a Potential Tool in Triple-Negative Breast Cancer Treatment” for publication in Cancers. Overall, we find Reviewer suggestions to be helpful, and constructive, and the corresponding revisions have strengthened the paper in multiple ways. According to your suggestions, we have expanded our manuscript with the content you suggested (the changes made in the manuscript are marked in red).
“This paper reports very interesting results with a new Selenium containing compound that seems to stimulate at very low concentrations also the apoptosis and autophagy of the MDA-MB-231 triple negative breast cancer cell lines.
It is great to see that this new compound works so well but of course there is no toxicity study and we can't know how it works in non-cancerogenic cells.
The authors mention the necessity of further studies. My suggestion is to emphasize more the possibility of high toxicity.”.
Response: Thank you very much for your favorable review. As suggested by the Reviewer, the possibility of high toxicity and the limitations associated with it have been added to the discussion along with a proposal to reduce it.
We hope that our answer will satisfy Reviewer and Editor. We thank you for consideration and await your decision.
